# Clinical NEC prevention practices drive different microbiome profiles and functional responses in the preterm intestine

Charlotte J. Neumann [1], Alexander Mahnert [1], Christina Kumpitsch [1], Raymond Kiu [2], Matthew J. Dalby [2], Magdalena Kujawska [3], Tobias Madl [4,5], Stefan Kurath-Koller[6], Berndt Urlesberger [7,8], Bernhard Resch [7,8] ✉, Lindsay J. Hall [2,3,9] & Christine Moissl-Eichinger [1,5] ✉

Preterm infants with very low birthweight are at serious risk for necrotizing enterocolitis. To functionally analyse the principles of three successful preventive NEC regimens, we characterize fecal samples of 55 infants (<1500 g, $n = 383$, female = 22) longitudinally (two weeks) with respect to gut microbiome profiles (bacteria, archaea, fungi, viruses; targeted 16S rRNA gene sequencing and shotgun metagenomics), microbial function, virulence factors, antibiotic resistances and metabolic profiles, including human milk oligosaccharides (HMOs) and short-chain fatty acids (German Registry of Clinical Trials, No.: DRKS00009290). Regimens including probiotic *Bifidobacterium longum* subsp. *infantis* NCDO 2203 supplementation affect microbiome development globally, pointing toward the genomic potential to convert HMOs. Engraftment of NCDO 2203 is associated with a substantial reduction of microbiome-associated antibiotic resistance as compared to regimens using probiotic *Lactobacillus rhamnosus* LCR 35 or no supplementation. Crucially, the beneficial effects of *Bifidobacterium longum* subsp. *infantis* NCDO 2203 supplementation depends on simultaneous feeding with HMOs. We demonstrate that preventive regimens have the highest impact on development and maturation of the gastrointestinal microbiome, enabling the establishment of a resilient microbial ecosystem that reduces pathogenic threats in at-risk preterm infants.

About eleven percent of all infants worldwide are born prematurely, i.e., before 37 weeks' gestation[1]. Very low birth weight (VLBW) preterm infants (<1500 g) are particularly vulnerable to acute and long-term clinical complications. Of particular concern is the development of necrotizing enterocolitis (NEC), a serious gastrointestinal threat that occurs in 7–11% of VLBW infants[2]. In such cases, mortality can reach up to 30%[3].

NEC is a devastating multifactorial disease that is driven in part by perturbations of the microbiome, including colonization and overgrowth of certain microbes with potentially pathogenic potential such as *Escherichia coli* or *Clostridium perfringens*[4].

Given the rapid onset of NEC, a number of neonatal intensive care units (NICUs) have developed specific NEC prophylaxis programmes that include the use of probiotics, antibiotics, and differentiated feeding protocols and that have resulted in a recent, substantial decrease in NEC rates in preterm infants[5].

Probiotic treatments are usually based on the use of *Bifidobacterium* and *Lactobacillus* species[6]. *Bifidobacterium*, in particular, is

considered as an important member of the resident infant microbiome that is maintained into early childhood and promotes healthy infant development[7,8].

Antibiotics are administered intravenously at the first signs of infection to control early-onset sepsis. As a result, the majority of VLBW infants are exposed to antibiotics in the first few days of life and for extended periods of time[9]. A number of publications have emphasized the need for responsible antibiotic usage in such vulnerable patients, as their use is associated with the risk of infection with multi-drug resistant (MDR) microorganisms[10] and is believed to have other, largely unknown, long-term effects. Overall, antibiotic exposure is often considered as preventable[9,11].

Additionally, the use of enteral antibiotics may be effective as NEC prophylaxis. The findings of the Cochrane Neonatal Collaborative Review Group[12] suggest that oral administration of prophylactic enteral antibiotics results in a statistically significant reduction in NEC and in NEC-related deaths in low-birth-weight preterm infants. However, the risks of enteral antibiotics have not yet been quantified; thus, this strategy has never been widely adopted due to concerns about the emergence of resistant bacteria and the absorption of antibiotics from the gut[13]. However, such adverse effects have not been reported so far[14].

Human milk (HM), the gold standard for infant feeding, is a surprisingly complex synbiotic that contains probiotic bacteria and prebiotics to nourish probiotic bacteria. Prebiotic human milk oligosaccharides (HMOs) are complex carbohydrates present in large quantities in HM that are not broken down by intestinal enzymes. Therefore, they serve only as a specific substrate for certain bacteria in the infant's gastrointestinal tract (GIT), such as mainly *Bifidobacterium* (*Bifidobacterium bifidum* and *Bifidobacterium longum* subsp. *infantis*) but also *Bacteroides* (*Bacteroides vulgatus* and *Bacteroides fragilis*)[15,16]. Indeed, *Bifidobacterium* is enriched in infants fed HM[17], due to its ability to metabolize HMOs. The sophisticated, individual complexity of HM can only be partially mimicked by formula milk (FM); nevertheless, newer products also contain standardized pre- and probiotics for optimal nutrition.

Southern Austrian neonatal units have implemented various combinations of these prophylactic measures with great success, resulting in an exceptionally low average NEC rate of 2.9% in VLBW infants (2007–2016[18]).

In this work, we take the opportunity to deeply analyse the mechanism for success across these different regimens on the level of the gut microbiome and metabolome.

We recruited 55 VLBW infants in three closely neighboured hospital centers (Graz, Klagenfurt, Leoben), that differ in antibiotic treatment (enteral gentamicin or none), antifungal treatment (enteral nystatin or parenteral fluconazole), probiotic use (*Lactobacillus rhamnosus* LCR 35, Bifidobacterium *longum* subsp. *infantis* NCDO 2203 in combination with *Lactobacillus acidophilus NCDO 1748*, or none) and feeding (HM, FM). Using a multi-omics approach, we examine the composition and function of the microbiome and its metabolites in the first weeks of life to understand the importance of the interactions among dietary components, antibiotics and probiotics.

Our study differs from previous studies in that a focus is placed on different NEC-prevention protocols, not in just one but in three different clinics. This study setup also allowed us to avoid the problematic cross-contamination of probiotics into the control groups[19,20]. To understand the effects and mechanisms of the different treatments, we analyse the microbiome on a multi-kingdom level and include functional metagenomics and metabolomics, as well as genome profiling on the species level. We conclude our study with a suggestion to further improve existing protocols to support a healthy microbiome development in VLBW infants by combining effective probiotics (including *Bifidobacterium longum* subsp. *infantis* NCDO 2203) and human milk.

## Results

Details on the study design are provided within the Methods. In brief, fecal samples were collected prospectively in three independent NICUs in Austria using a different NEC prophylaxis regimen (Table 1, in Methods) from preterm infants with a birthweight <1500 g. Samples were collected every other day, starting with the meconium, up until two weeks of age. The study groups do not differ statistically significantly in any observed metadata except for the length of hospital stay of the mothers after birth[21], Table 2 therein).

It should be mentioned that the three clinical situations studied here differed in a number of confounding factors, both recorded and possibly unrecorded, and we can only draw conclusions based on the overall setting in each NICU, which includes medication and probiotic regimens. However, the three hospitals are geographically very close to each other, so the patient catchment area also overlaps, and we may consider other factors to be minor compared with the medication and feeding protocols. Indeed, PERMANOVA analyses revealed that the combined variables *Bifidobacterium* administration (yes/no), feeding protocol (formula milk, FM; human milk, HM; mixed), gentamycin administration (yes/no) had the same or even stronger effect ($R^2 = 0.6763$ (time point 7), $R^2 = 0.3430$ (tp3), $R^2 = 0.2710$ (all); $p = 0.001$ (all tests)), than the grouping according to the hospital ($R^2 = 0.6763$ (time point 7), $R^2 = 0.2618$ (time point 3), $R^2 = 0.2623$ (all); $p = 0.001$ (all tests)), indicating that the major observed differences were indeed driven by the regimens (Suppl. Table 1, amplicon dataset).

### Early-life therapy regimen influences microbiome composition and development across all microbial domains
Assessing the microbial composition of all infants with metagenomic analyses and 16S rRNA gene sequencing, we detected an association of

**Table 1 | Prophylactic regimens of probiotics, antibiotics, antifungals and feeding protocols of the three different neonatal intensive care units (NICUs): Graz (G), Klagenfurt (K), and Leoben (L)**

| | NICU Graz (G) | NICU Klagenfurt (K) | NICU Leoben (L) |
|---|---|---|---|
| Probiotics | "Antibiophilus" *Lactobacillus rhamnosus* LCR 35 $1 \times 10^9$ CFU/d, oral., split into 2 doses per day | "Infloran" *Bifidobacterium longum* subsp. *infantis* NCDO 2203 $2 \times 10^9$ CFU/d *Lactobacillus acidophilus* NCDO 1748 $2 \times 10^9$ CFU/d in combination, oral | None |
| Antibiotics | Gentamicin 7 mg/kg, every 12 h, oral | None | Gentamicin 7 mg/kg, every 12 h, oral |
| Antifungal agents | Nystatin 10,000 U/kg every 6 h, oral | Fluconazole 6 mg/kg, every 72 h (<1000 g BW), intravenous | Nystatin 10,000 U/kg every 6 h, oral |
| Feeding Protocol | HM favored over FM | FM and in few cases additionally pasteurized HM | HM favored over FM |

*CFU* Colony forming units, *HM* Human milk, *FM* Formula milk.

**Table 2 | Distribution of overall metagenomic reads across the domains of life and between the centers on different taxonomic levels**

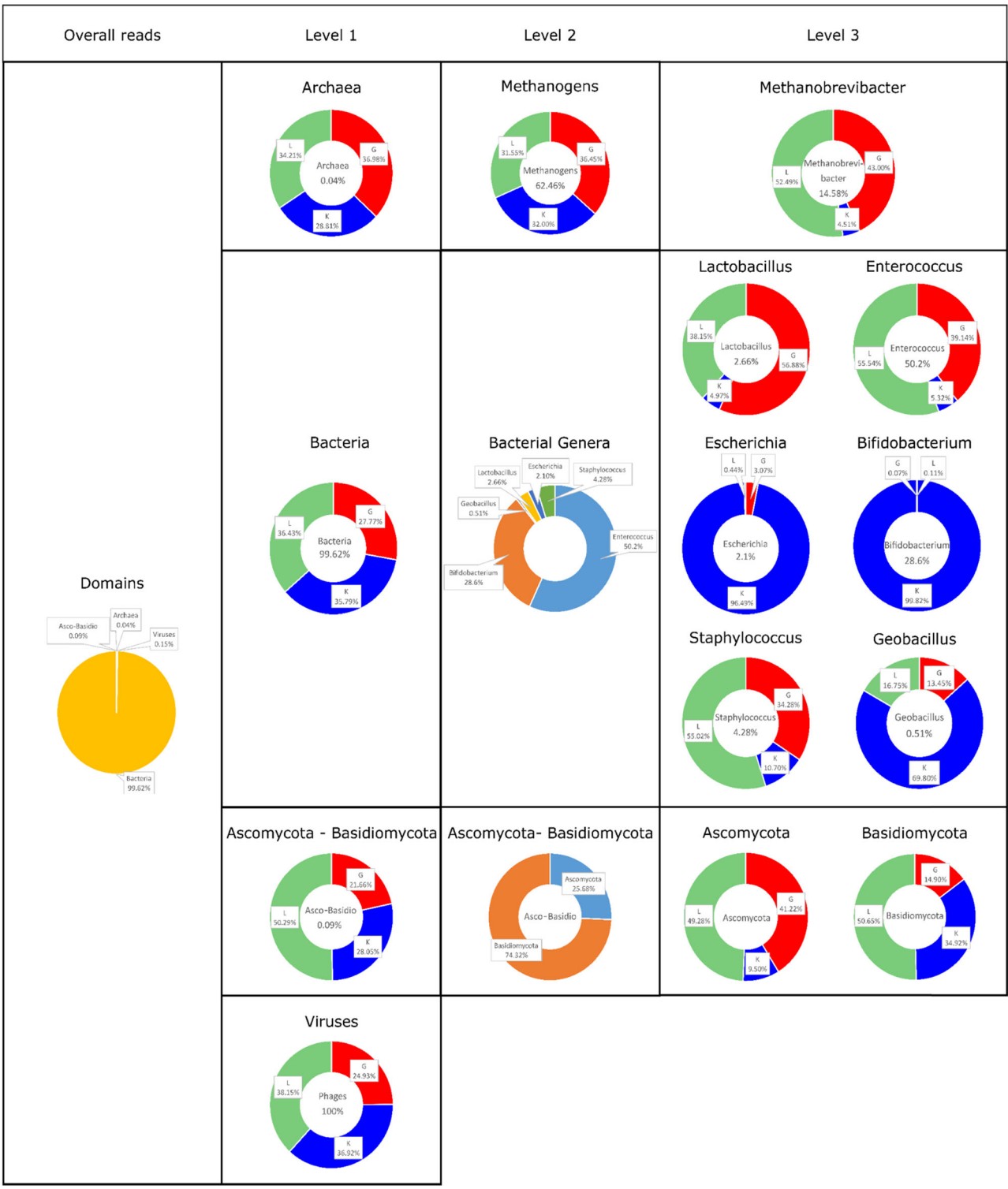

the preventive NEC regimen on all microbial domains and groups, including bacteria (99.62% of all metagenomic reads), their phages and viruses (0.15%), but also on archaea (0.04%) and fungi (Ascomycota/ Basidiomycota: 0.09%) (Table 2, Fig. 1, Fig. 2).

The role or even presence of archaeal signatures in the premature infants' gut is still unclear and underexplored. Previous publications

concluded that infants generally do not carry a substantial amount of archaea until five years of age[22], and especially in preterm infants, archaea were found only in a little proportion of screened infants[23,24]. Our archaea-focused approach enabled us to successfully detect 290 different ASVs with amplicon-based analyses and 75 different archaeal species with metagenomic-based sequencing. In particular,

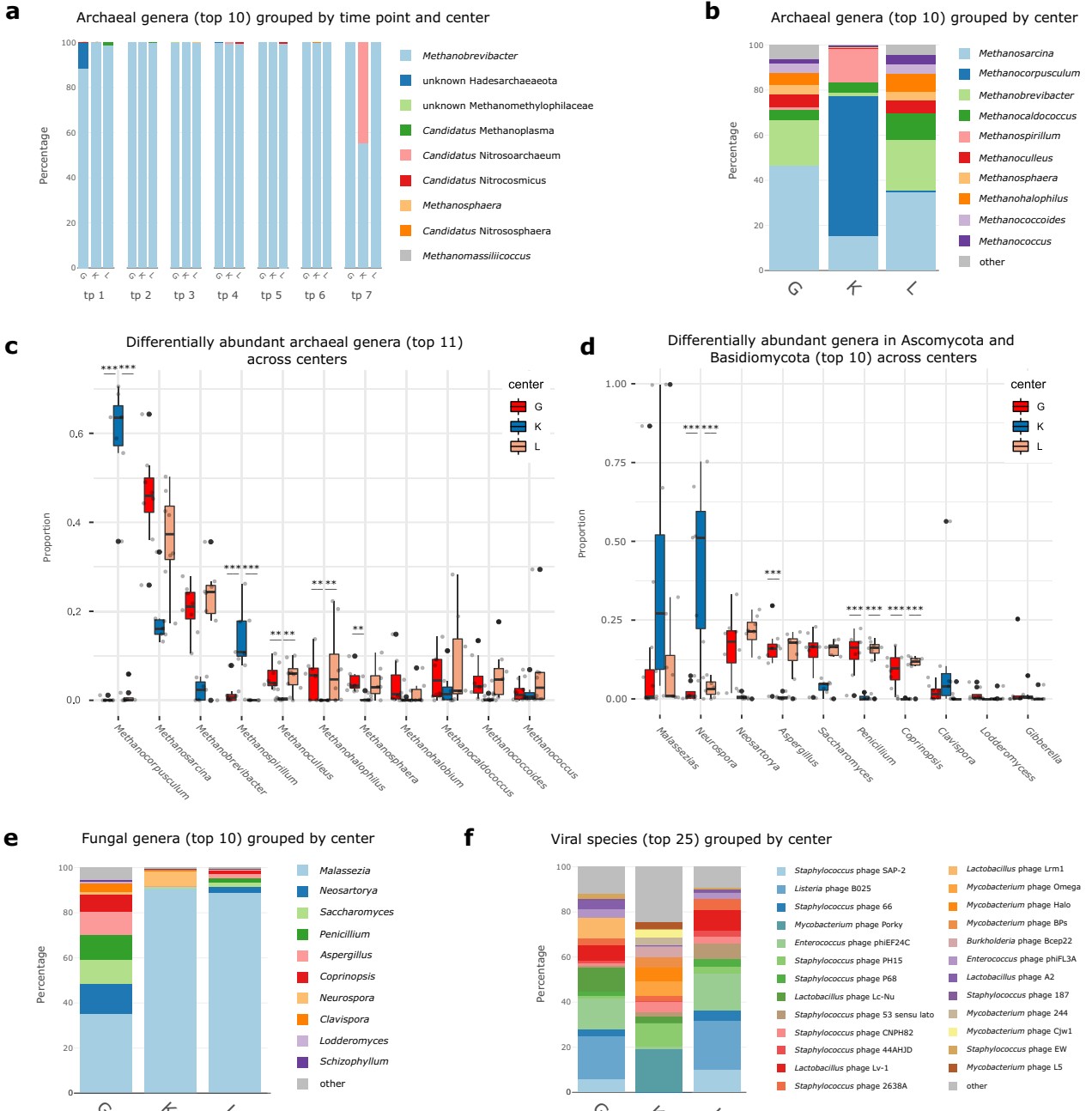

**Fig. 1 | Overall distribution and abundances of microbial signatures of different domains, according to the different centers. a** Stacked bar plot of relative abundances of the top ten archaeal genera in the amplicon data set, displayed per center at time points (tp) tp1–7. **b** Stacked bar plot of the top ten relative abundances of methanogenic archaeal genera in the MGS (metagenomic) dataset for tp7 per center. **c** Box plot of relative abundances of the top eleven methanogenic archaeal genera per center (MGS data, tp7, $n = 23$ biologically independent samples, DSeq2): *Methanocorpusculum*: K:G $q < 0.001$, K:L $q < 0.001$, G:L $q = 0.945$; *Methanosarcina*: K:G $q = 0.261$, K:L $q = 0.054$, G:L $q = 0.945$; *Methanobrevibacter*: K:G $q = 0.052$, K:L $q = 0.138$, G:L $q = 0.945$; *Methanospirillum*: K:G $q < 0.001$, K:L $q < 0,001$, G:L $q = 0.192$; *Methanoculleus*: K:G $q < 0.001$, K:L $q = 0.009$, G:L $q = 0.945$; *Methanosphaera*: K:G $q = 0.028$, K:L $q = 0.138$, G:L $q = 0.945$; *Methanohalobium*: K:G $q = 0.052$, K:L $q = 0.774$, G:L $q = 0.945$; *Methanocaldococcus*: K:G $q = 0.237$, K:L $q = 0.576$, G:L $q = 0.945$; *Methanococcoides*: K:G $q = 0.112$, K:L $q = 0.264$, G:L $q = 0.945$; *Methanococcus*: K:G $q = 0.371$, K:L $q = 0.964$, G:L $q = 0.945$. **d** Box plot of relative abundances of the top ten genera of Ascomycota and Basidiomycota per center (MGS data, tp7, $n = 23$ biologically independent samples, DSeq2): *Malassezia*: K:G $q = 0.021$, K:L $q = 0.095$, G:L $q = 0.792$; *Neurospora*: K:G $q < 0.001$, K:L

$q = 0.001$, G:L $q = 0.856$; *Neosartorya*: K:G $q < 0.001$, K:L $q < 0.001$, G:L $q = 0.995$; *Aspergillus*: K:G $q < 0.001$, K:L $q < 0.001$, G:L $q = 0.995$; *Saccharomyces*: K:G $q = 0.289$, K:L $q = 0.035$, G:L $q = 0.995$; *Penicillium*: K:G $q < 0.001$, K:L $q < 0.001$, G:L $q = 0.995$; *Coprinopsis*: K:G $q < 0.001$, K:L $q < 0.001$, G:L $q = 0.995$; *Clavispora*: K:G $q = 0.293$, K:L $q = 0.04$, G:L $q = 0.291$; *Lodderomyces*: K:G $q = 0.056$, K:L $q = 0.834$, G:L $q = 0.006$; *Gibberella*: K:G $q = 0.056$, K:L $q = 0.785$ 0.04, G:L $q = 0.785$. **e** Stacked bar plot of the relative abundances of the top ten genera of Ascomycota and Basidiomycota in the MGS dataset for tp7 per center. **f** Stacked bar plot of the top 25 relative abundances of phage species in the MGS dataset for tp7 per center. Significance levels are indicated with asterisks for $q < 0.001$ (***), $q < 0.01$ (**), $q < 0.05$ (*) for differentially abundance testing by DSeq2, adjusted for multiple comparisons. Centers are abbreviated by G (Graz), L (Leoben) and K (Klagenfurt). For boxplots, the upper, middle and lower horizontal lines of the box represent the upper, median and lower quartile; their whiskers depict the smallest or largest values within 1.5-fold of the interquartile range. Top genera/species were calculated across all samples. Source data are provided as a Source Data file (see Github repository).

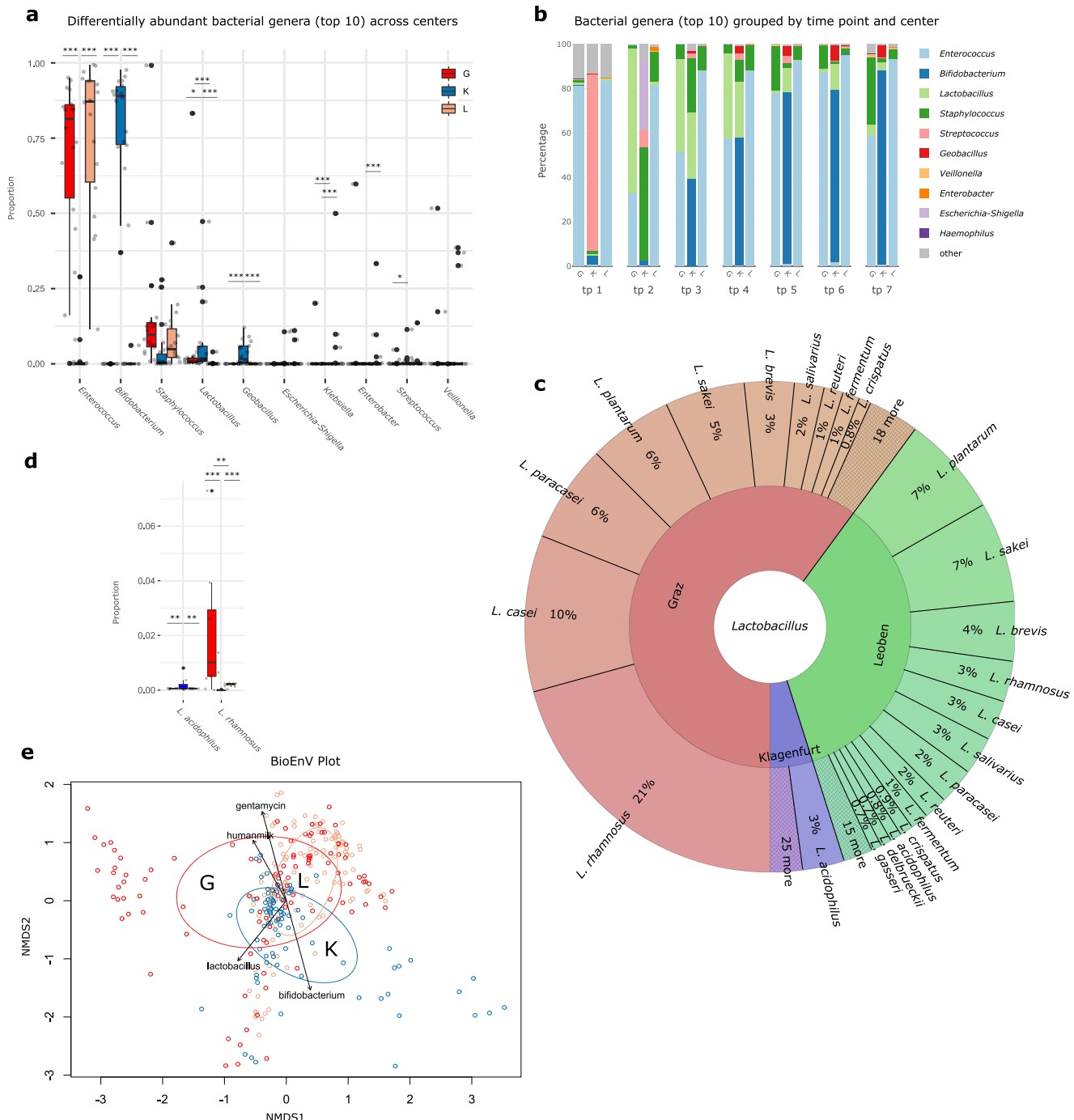

**Fig. 2 | Distribution of bacterial taxa between the centers and influence of probiotic species. a** Box plot of relative abundances of the top ten bacterial genera per center (amplicon data, tp7, *n* = 23 biologically independent samples, DSeq2) *Enterococcus*: K:G *q* < 0.001, K:L *q* < 0.001, G:L *q* = 1; *Bifidobacterium*: K:G *q* < 0.001, K:L *q* < 0.001, G:L *q* = 1; *Staphylococcus*: K:G *q* = 0.642, K:L *q* = 1, G:L *q* = 0.274; *Lactobacillus*: K:G *q* = 0.034, K:L *q* < 0.001, G:L *q* < 0.001; *Geobacillus*: K:G *q* < 0.001, K:L *q* < 0.001, G:L *q* = 1; *Escherichia-Shigella*: K:G *q* = 1, K:L *q* = 0.979, G:L *q* = 1; *Klebsiella*: K:G *q* = 1, K:L *q* < 0.001, G:L *q* < 0.001; *Enterobacter*: K:G *q* = 0.916, K:L *q* < 0.001, G:L *q* = 1; *Streptococcus*: K:G *q* = 0.018, K:L *q* = 0.443, G:L *q* = 0.765; *Veillonella*: K:G *q* = 0.050, K:L *q* = 0.111, G:L *q* = 0.768; **b** Stacked bar plot of relative abundances of top ten bacterial genera in the amplicon data set per center each at time points tp1–7. **c** Krona chart of the distribution of species of the *Lactobacillus* genus between the centers (MGS data, tp7). **d** Log percentages of relative abundance of probiotically administered *Lactobacillus* genera (MGS data, tp7, *n* = 23 biologically independent samples, DSeq2) *Lactobacillus acidophilus*: K:G *q* = 0.003, K:L *q* = 0.002, G:L *q* = 1; *Lactobacillus rhamnosus*: K:G *q* < 0.001, K:L *q* = 0.001, G:L *q* = 0.001 (i) *L. acidophilus* NCDO 1748 and (ii) *L. rhamnosus* LCR 35. **e** Biplot of BioEnv with correlations of the Euclidean distances for the metadata of dissimilarities between the centers (administration of gentamicin, of probiotic *Lactobacillus* or *Bifidobacterium* and human milk), amplicon data. G in red, K in blue, L in green; significance levels are indicated with asterisks for *q* < 0.001 (***), *q* < 0.01 (**), *q* < 0.05 (*) for differentially abundance testing by DSeq2, adjusted for multiple comparisons. Centers are abbreviated by G (Graz), L (Leoben), and K (Klagenfurt). For boxplots, the upper, middle and lower horizontal lines of the box represent the upper, median, and lower quartile; their whiskers depict the smallest or largest values within 1.5-fold of the interquartile range. Source data are provided as a Source Data file (see Github repository).

*Methanobrevibacter* was abundant, as it was detected in all infants in at least one sample and across all time points. In addition to typical human-associated archaea, *Methanobrevibacter*, *Methanosphaera*, Methanomethylophilaceae (incl. Methanomassiliicoccus), and various Nitrososphaeria (likely derived from skin sources)[25,26] (Fig. 1a, amplicon data), abundant signatures of *Methanosarcina* and *Methanocorpusculum* were additionally identified using shotgun metagenomics (Fig. 1b, MGS data), indicating that even VLWB newborns are in contact with a wide diversity of archaea. Archaea reflected the center, with, for example, *Methanocorpusculum* and *Methanospirillum* being significantly more abundant in samples from Klagenfurt (K) (*Methanocorpusculum*, DESeq2, K:G $q < 0.001$, K:L $q < 0.001$), than in Graz (G) and Leoben (L) (Fig. 1c, MGS data).

The contribution of fungal signatures to the overall microbiome was largely limited (also probably due to the application of antifungals) to Basidiomycota and Ascomycota (Fig. 1d, e, MGS data), which, however, also revealed a center-specific pattern: *Neosarorya*, *Penicillum*, *Aspergillus* and *Coprinopsis* (significantly increased in G and L with DESeq2 $q < 0.001$) were antiparallel to *Malassezia*, *Neurospora* and *Clavispora*, which were increased in K.

In order to confirm the center-specific profiles of the multiple-component microbiome data (further details, see below), a network analysis was performed based on the ten most differentially abundant genera of bacteria, methanogens, ascomycota/basidiomycota and phages (Suppl. Fig. 2 MGS data). The network revealed the formation of two separate clusters, whose nodes were mainly composed of K taxa and taxa from G and L, respectively. Those two clusters were connected with negative associations only, underlining the separating effect of the different regimens (see also PERMANOVA results, mentioned above). Some taxa were even shared by both centers (e.g., *Methanococcus*, *Candida*), suggesting an interaction between the domains.

## Supplemented *Bifidobacterium longum* subsp. *infantis* outweighs natural pathobiont colonizers and co-administered *Lactobacillus acidophilus*

The bacteriomes of the infants were mainly characterized by the predominance and differential abundance of six bacterial key taxa, namely *Enterococcus*, *Bifidobacterium*, *Lactobacillus*, *Staphylococcus*, *Geobacillus* and *Escherichia* (Fig. 2a, amplicon data, tp7).

*Enterococcus* was found to predominate the bacterial microbiome in G and L (-77%), followed by *Lactobacillus* and *Staphylococcus*, the relative abundance of which decreased with maturation. In contrast, the K samples were dominated by *Bifidobacterium* at each time point and reached a relative abundance of > 82% at tp7 (Fig. 2b, amplicon data). Thus, G and L were dominated by a typical colonizer of the GI in preterm infants, whereas K samples showed an overall predominance of a supplemented probiotic taxon. The phenomenon of detection of *Geobacillus* signatures exclusively in K samples has been discussed previously[21] and in the next section.

Notably, 16 of all 89 phage species were strongly associated with bacterial key species, resulting in a center-specific, strongly differing phage profile (Fig. 1f, MGS data). No bacteriophages were identified for *Bifidobacterium*, *Escherichia-Shigella* and *Geobacillus*; however, other phages from key species correlated with the relative abundance of their host, exemplified by *Streptococcus* in Suppl. Fig. 3. In particular, phages targeting *Lactobacillus* (Kruskal-Wallis; G:K $q = 0.004$, G:L $q = 0.003$) and *Enterococcus* (Kruskal-Wallis; K:L $q = 0.012$) were lowest in K.

Shotgun metagenomics confirmed the significantly differential abundance of *Bifidobacterium* in the three centers that was already observed in the amplicon sequencing approach (DESeq2, K:G $q < 0.001$, K:L $q < 0.001$). Although *B. longum* subsp. *infantis* NCDO 2203 was the only *Bifidobacterium* administered in K, nine additional *Bifidobacterium* species were detected, with six species present in all K

infants (*B. longum*, *B. dentium*, *B. breve*, *B. bifidum*, *B. animalis*, *B. adolescentis*). *B. longum* accounted for 95% of all reads from *Bifidobacterium*, and indeed this taxon was verified as *Bifidobacterium longum* subsp. *infantis* NCDO 2203 by genomic comparisons (see Material and Methods and Suppl. Table 3b). These analysis results support our assumption that the major *Bifidobacterium* signatures in K infants were indeed from the administered probiotic. Of note, the signatures of *B. longum* subsp. *infantis* are abbreviated as *B. infantis* in the following.

Naturally, bifidobacteria are uncommon in the premature infants' GI in the first days of life and start colonizing naturally beginning from week four and on[6,27]. Our data also indicate an absence of bifidobacteria in preterm infants who did not receive it via supplementation. Furthermore, administration of *B. infantis* NCDO 2203 resulted in very high abundances over other bacteria, including the co-administered *L. acidophilus* NCDO 1748: Although both probiotic species, *B. infantis* NCDO 2203 and *L. acidophilus* NCDO 1748, were administered in equal amounts in K, *L. acidophilus* NCDO 1748 was substantially lower in abundance than the predominant *B. infantis* NCDO 2203 (0.21% relative abundance of *L. acidophilus* NCDO 1748 at tp7 vs. 75.69% relative abundance of *B. infantis* NCDO 2203). We suggest, that the colonizing potential of *B. infantis* NCDO 2203 is higher than the one of *L. acidophilus* NCDO 1748 probably due to its different metabolic capacities. In agreement with our data, it was already shown before, that lactobacilli do not colonise the preterm gut in large numbers[28].

## Probiotic administration of lactobacilli increases their natural abundance and diversity

In total, 27 species of the genus *Lactobacillus* (see details on classification issues in Methods), were detected by shotgun sequencing with different profiles between the centers (Fig. 2c). Probiotic lactobacilli were administered only in K (*L. acidophilus* NCDO 1748) and G (*L. rhamnosus* LCR 35). The presence of these lactobacilli in the infants' intestinal samples was confirmed by amplicon sequencing but also by read-mapping (Fig. 2d, Suppl. Table 3a).

*Lactobacillus rhamnosus* was also detected with low relative abundance (3%) in L, where it was not administered, suggesting that this species is a low abundant part of the natural infant gut microbiome of preterm infants and is possibly transmitted through breastfeeding or other sources[29]. In G, *L. rhamnosus* LCR 35 exhibited the highest abundance in G infants (21%) indicating a seven-fold increase in probiotic lactobacilli through its probiotic administration. K had the lowest absolute *Lactobacillus* abundance, with 56% of all *Lactobacillus* reads representing *L. acidophilus* NCDO 1748, the species administered there (K: 45,268 reads; G: 10,207; L: 12,431). Similar to *L. rhamnosus*, *L. acidophilus* was also detected in almost all infants in all centers, even when not supplemented as probiotics (Fig. 2d).

Although *L. acidophilus* NCDO 1748 was administered in K, it does not result in high abundances, compared to *B. infantis* NCDO 2203 administered at the same concentration. This is reflected by an *L. acidophilus* NCDO 1748 / *B. infantis* NCDO 2203 ratio of -1:300 and *Lactobacillus*: *Bifidobacterium* ratio of -1:200 (for genera see Suppl. Table 4).

PERMANOVA analyses confirmed the superior impact of *Bifidobacterium* ($R^2 = 0.2084$ (all samples), $R^2 = 0.1258$ (tp 3), $R^2 = 0.5716$ (tp7)) over *Lactobacillus* administration ($R^2 = 0.04733$ (all samples), $R^2 = 0.0631$, $R^2 = 0.1394$), explaining up to 57.16% and 13.94% of the observed variance, respectively (Suppl. Table 1). The dominance of *Bifidobacterium* over *Lactobacillus* could also be underlined with the BioEnv Biplot (Fig. 2e, amplicon data). This plot shows the metadata whose Euclidean distances have the maximum (rank) correlation with community dissimilarities. In particular, the administration of *Bifidobacterium* and gentamicin correlated strongly with dissimilarities between the centers, whereas the administration of lactobacilli and HM correlates to a lesser degree.

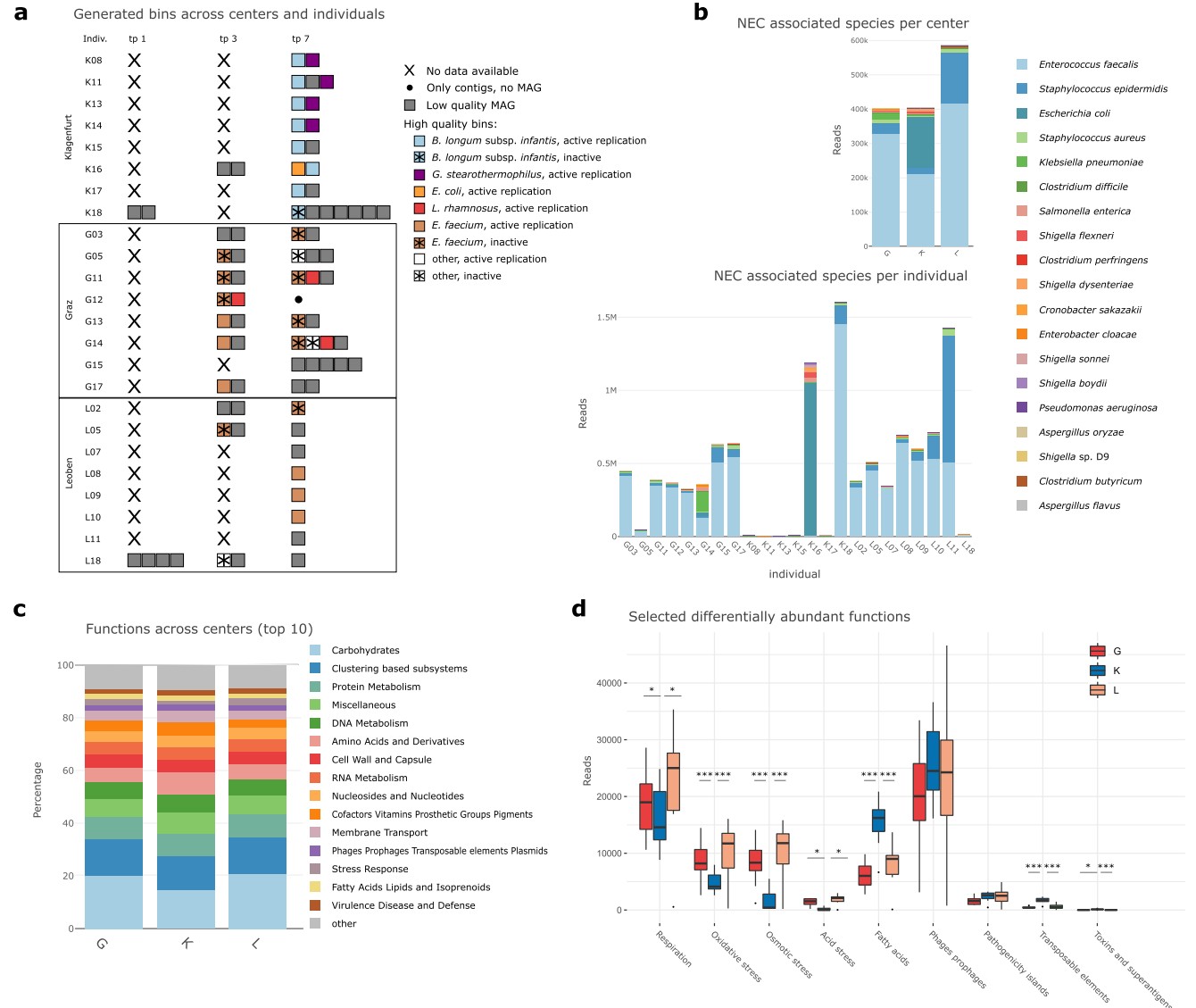

**Fig. 3 | Replication values for MAGs, distribution of potentially NEC causing microbes, and specific functions. a** Retrieved MAGs per center, individual and time points tp1, tp3 and tp7: availability, quality, iRep replication status [min: 1.348, max: 1.998, mean: 1.654] and taxonomic placement; **b** Read numbers of microbial species that were correlated with NEC (Necrotizing Enterocolitis) previously, per center and per individual, MGS data. **c** Stacked bar plot of relative abundances of top fifteen microbiome functions per center, MGS data; **d** reads of ten selected differentially abundant functions per center at tp7, MGS data, $n = 23$ biologically independent samples: respiration: K:G $q = 0.007$, K:L $q = 0.014$, G:L $q = 0.703$; oxidative stress: K:G $q < 0.001$, K:L $q < 0.001$, G:L $q = 0.949$; osmotic stress: K:G $q < 0.001$, K:L $q < 0.001$, G:L $q = 0.949$; acid stress: K:G $q = 0.033$, K:L $q = 0.029$, G:L

$q = 0.949$; fatty acids: K:G $q < 0.001$, K:L $q < 0.001$, G:L $q = 0.949$; phages prophages: K:G $q = 0.997$, K:L $q = 0.640$, G:L $q = 0.949$; pathogenicity islands: K:G $q = 0.859$, K:L $q = 0.305$, G:L $q = 0.849$; transposable elements: K:G $q < 0.001$, K:L $q < 0.001$, G:L $q = 0.949$; toxins and superantigens: K:G $q = 0.005$, K:L $q < 0.001$, G:L $q = 0.760$;. G in red, K in blue, L in green; significance levels are indicated with asterisks for $q < 0.001$ (***), $q < 0.01$ (**), $q < 0.05$ (*) for differentially abundance testing by DSeq2, adjusted for multiple comparisons. Centers are abbreviated by G (Graz), L (Leoben), and K (Klagenfurt). For boxplots, the upper, middle, and lower horizontal lines of the box represent the upper, median, and lower quartile; their whiskers depict the smallest or largest values within 1.5-fold of the interquartile range. Source data are provided as a Source Data file (see Github repository).

## Key species are reflected by MAGs and active replication of probiotic species could be inferred

Samples from tp3 (days 5–8) and tp7 (days 13–21) allowed for deep metagenomic sequencing and genome binning. Metagenome assembled genomes (MAGs) were obtained for *B. infantis* NCDO 2203 (K), *G. stearothermophilus* (K), *E. faecium* (G, L), *E. coli* (K), *L. rhamnosus* (G), *Veillonella parvula* (G, inactive), *Klebsiella oxytoca* (G, inactive) and *Escherichia flexneri* (L, inactive). Successful binning of the bacterial genomes followed the scheme of probiotic supplementation, with *B. infantis* NCDO 2203 MAGs in K samples and *L. rhamnosus* LCR 35 MAGs in G samples. Of the L samples, only MAGs of *Enterococcus faecium* were obtained (Fig. 3a, MGS data). Application of iRep[30] suggested that MAGs might correspond to actively replicating bacteria (iRep values

>1), and in consequence indicate niche colonization by probiotic (*B. infantis* NCDO 2203 and *L. rhamnosus* LCR 35) or naturally predominant bacteria (*E. faecium*) (Fig. 3a, MGS data).

Notably, high-quality *G. stearothermophilus* genomes with iRep values above 1.41 were isolated from four out of eight infant samples from K (tp7) (see also ref. [21]). *G. stearothermophilus* is a frequent, most probably harmless contaminant in milk plants[31], which probably transforms to an active form during formula milk preparation and is then ingested by the infant. Thermophilic *G. stearothermophilus* grows in the temperature range of 40–70 °C, with optimal growth rates achieved at 55–65 °C[32,33]; thus, an active proliferation in the infant GI is unlikely. *Staphylococcus* MAGs could not be retrieved, despite its high abundance and identification as a key microorganism in this study (Fig. 3a).

## Low level occurrence of diverse types of potentially pathogenic bacteria

Next, we searched specifically for signatures that had been associated with outbreaks or cases of NEC in previous reports[27]. In our shotgun metagenomic dataset, we identified *E. faecalis* as having the highest abundance in this dataset, while *S. epidermidis*, *E. coli* and others were found in varying amounts in samples from tp7 (Fig. 3b). While the infants from the other centers showed a more or less homogenous presence of potential NEC-causing microorganisms, the hits in K samples concentrated solely on two infants, K16 and K18 (Fig. 3b). Reads of *E. faecalis* derived mainly from K18, whereas a high number of *E. coli* signatures originated almost exclusively from K16, the only infant in this subset that developed NEC later on. Furthermore, a MAG of probably active *E. coli* could be obtained from this infant at tp7 (i.e., 14 days after birth), suggesting the possible initial bloom of *E. coli* already at this early time point before NEC is usually diagnosed. As *E. coli* MAGs were not retrieved from any other sample, our data support the potential for microbiome analyses to be used for NEC monitoring and diagnosis (Fig. 3a).

## Functional profiles possibly mirror earlier gut maturity in infants following regiments with *B. infantis*

We also profiled microbiome functional characteristics at tp7 given the sufficient sequencing coverage obtained at this time point (Fig. 3c, MGS data). K samples showed an overall significantly reduced level of genes involved in osmotic stress, acid stress and respiration (DESeq2, osmotic stress, K:G $q < 0.001$, K:L $q < 0.001$; acid stress; K:G $q = 0.033$, K:L $q = 0.029$; respiration; K:G $q = 0.007$, K:L $q = 0.014$). Additionally, the GI microbiome in K was capable to degrade more complex sugars reflected by the increased relative abundance of genes involved in polysaccharide metabolism, which was also confirmed by metabolomics (Fig. 3d; see also below). Summing up those points, it seems as if already rather anaerobic and more complex metabolism takes place in K than in G and L. In general, complex metabolism and anoxic environment are characteristics of a more mature gut microbiome.

Notably, the number of genes related to transposable elements (involved in the distribution of pathogenic genomic features), were significantly higher in K samples (Kruskal-Wallis, K:G $q < 0.001$, K:L $q < 0.001$), suggesting a potentially higher bacterial pathogenesis signature in these infants.

Next, we performed NMR-based metabolomics on 111 samples from tp1, tp3 and tp7 to determine the concentration of short-chain fatty acids (SCFAs) and complex sugars in the infants' stool samples. We found a general increase in acetic acid, formic acid, valeric acid and butyric acid over time in all centers, regardless of the microbiome composition or probiotic supplementation (Suppl. Fig. 4a–d). However, unlike previous reports, we observed an unexpected spike in propionic acid at tp3 in all centers (Suppl. Fig. 4e)[34]. We hypothesize that this spike in propionic acid is related to a delayed uptake of propionate by the intestinal epithelium during maturation.

## Human milk supports *Bifidobacterium* by associated HMO conversion which is impaired by formula milk feeding

Genes involved in carbohydrate metabolism were the most abundant functional features in our shotgun metagenomic dataset. Notably, the samples from K had a significantly lower proportion than the other centers (Kruskal-Wallis, K:G $q = 0.005$, K:L $q = 0,026$). Upon further examination, particularly genes involved in the metabolism of monosaccharides were significantly lower in K than in L (Kruskal-Wallis, $q < 0.001$) (Fig. 4a). The significantly lower availability of monosaccharides in K was confirmed by a representative, metabolomic-based quantitative assessment of glucose and fructose. However, an overall increase of both compounds was observed in all centers over time (Suppl. Fig 4f, g). Similarly, genes involved in metabolism of di- and oligosaccharides were reduced in K samples, but not significantly

(Fig. 4a). In contrast, the gene proportion involved in polysaccharide metabolism was found to be significantly increased in K samples (Kuskal–Wallis-test, K:G $q < 0.001$, K:L $q < 0.001$) (Fig. 4a), indicating a higher genetic potential for the metabolism of complex sugars in K.

To answer the question regarding the genetic potential for complex HMO degradation, we searched for HMO gene clusters in the obtained MAGs and contigs. Indeed, the potential for HMO metabolism was notably higher in samples from K (Fig. 5d). The total hits with HMO gene clusters were G: 45, K: 307, L: 0, indicating a seven-fold higher numbers of HMO genes in K than in G. Moreover, only one MAG from G17 possessed the potential to convert and digest HMOs, in contrast to all infants from K with at least one MAG (Fig. 5d).

Using metabolomics, we assessed HMOs in the preterm stool samples. A total of 13 HMOs were detected measuring 111 stool samples at tp1, tp3 and tp7 (Fig. 4b). For most fucosylated HMOs (2′-fucosyllactose [2′FL], 3′fucosyllactose [3′FL], lacto-N-ducopentaoise I [LNFP1], lacto-N-fucopentaose III [LNFP3], lacto-N-difuco-hexaose [LNDFH], lactodifucotetraose [LDFT]) as well as LS-tetrasaccharide a [LSTa] and lacto-N-tetraose [LNT], at time point 7, a significantly decreased amount for K samples was detected as compared to G and L (Kruskal-Wallis; 2′FL, K:G $q = 0.091$, K:L $q = 0.033$; 3′FL, K:G $q = 0.062$, K:L $q = 0.020$; LDFP3, K:G $q = 0.003$, K:L $q = 0.009$; LNDFH, K:G $q = 0.059$, K:L $q = 0.003$; LDFT, K:G $q = 0.091$, K:L $q = 0.003$; LSTa, K:G $q = 0.003$, K:L $q = 0.003$; LSTb, K:G $q = 0.062$, K:L $q = 0.020$) (Fig. 4ci). For syalised HMOs (such as 3′-sialyllactose [3′SL], 6′-sialyllactose [6′SL], LS-tetrasaccharide b [LSTb], LS-tetrasaccharide c [LSTc], but also lacto-N-neotetraose [LNnT]), few differences between centers or time points were observed (Fig. 4cii).

We conclude that the K microbiomes have the largest potential to degrade HMOs effectively; however, this process is impaired by the lowered availability of HMOs by preferred formula feeding.

## Regimens and their key taxa correlate with crucial metabolites, antibiotic resistance genes, and virulence factors

We were interested in the correlations among the six microbial key players with HMOs, carbohydrates, amino acids and short-chain fatty acids, as measured using metabolomics (Fig. 5a) (additional metabolites are shown in Suppl. Fig. 5). *Bifidobacterium* (K) showed positive correlations with several carbohydrates (galactose and fructose), and weaker ones, with certain amino acids and short-chain fatty acids. It appears under HMO-depleted conditions, *Bifidobacterium* uses galactose and fructose as alternative substrates for its carbohydrate metabolism, producing formate and acetate. In the absence of *Bifidobacterium* (G and L), the role of formate and acetate production is taken over by *Enterococcus*, indicating its importance for early SCFA production, probably from citrate or pyruvate[35]. *Staphylococcus* showed an inconsistent pattern across the centers, supporting our hypothesis that this microorganism played a smaller role for the preterm GIT. The metabolic pattern of *Lactobacillus* (G, K) could not be clearly resolved, as lactate was not included in the metabolomic approach.

The antibiotic gentamicin (used in G and L; Table 1) has a broad spectrum of activity including: *Enterobacter*, *Escherichia*, *Proteus*, *Klebsiella*, *Pseudomonas*, *Serratia* and *Staphylococcus*[36]. Surprisingly, the influence of gentamicin on these genera was not evident in the preterm microbiome samples, and only limited perturbations were observed. However, it should be noted that, in this study, gentamicin was administered enterally and not intravenously, which may result in an altered mode of action. Acquired resistance to the gentamicin administered was rarely detected.

Overall, clear differences were observed in the antibiotic resistance (AMR) profiles between the key genera (Fig. 5b) and centers (Fig. 5c). AMR potential of the MAGs was substantially reduced in K samples as compared to samples from G and L (G: 93 hits, K: 35 hits, L: 123 hits), once again highlighting the influence of *Bifidobacterium* on

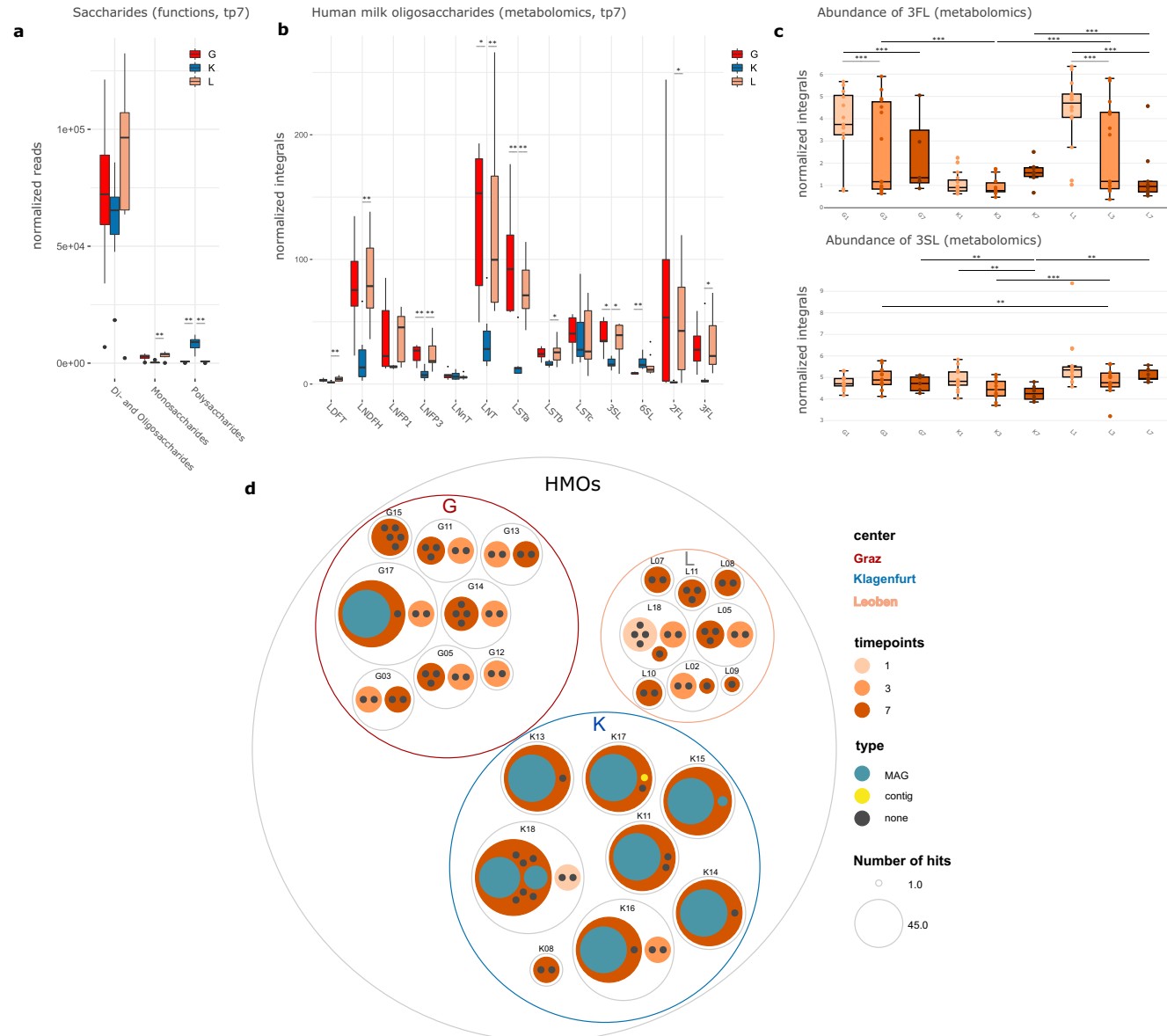

**Fig. 4 | Distribution patterns of sugars and HMOs. a** Differentially abundant functions (metagenomic dataset) associated with saccharides (di- and oligosaccharides, monosaccharides and polysaccharides) between the centers, tp7, $n = 23$; Monosaccharides: K:G $q < 0.001$, K:L $q < 0.001$, G:L $q = 0.949$; Di- and Oligosaccharides: K:G $q = 0.057$, K:L $q < 0.001$, G:L $q = 0.949$; Polysaccharides: K:G $q < 0.001$, K:L $q < 0.001$, G:L $q = 0.949$; **b** differentially abundant Human Milk Oligosaccharides (HMOs) between centers (metabolomic dataset, tp7, $n = 23$ biologically independent samples) LDFT: K:G $q = 0.091$, K:L $q = 0.003$, G:L $q = 1$; LNDFH: K:G $q = 0.059$, K:L $q = 0.003$, G:L $q = 1$; LNFP1: K:G $q = 1$, K:L $q = 1$, G:L $q = 1$; LNFP3: K:G $q = 0.003$, K:L $q = 0.009$, G:L $q = 1$; LNnT: K:G $q = 1$, K:L $q = 1$, G:L $q = 1$; LNT: K:G $q = 0.016$, K:L $q = 0.005$, G:L $q = 1$; LSTa: K:G $q = 0.003$, K:L $q = 0.003$, G:L $q = 1$; LSTb: K:G $q = 0.062$, K:L $q = 0.020$, G:L $q = 1$; LSTc: K:G $q = 1$, K:L $q = 1$, G:L $q = 1$; 3′SL: K:G $q = 0.019$, K:L $q = 0.018$, G:L $q = 1$; 6′SL: K:G $q = 0.003$, K:L $q = 0.413$, G:L $q = 0.088$; 2′FL: K:G $q = 0.091$, K:L $q = 0.033$, G:L $q = 1$; 3′FL: K:G $q = 0.062$, K:L $q = 0.020$, G:L $q = 1$; **c** differential abundance of the HMOs 3′-fucosyllactose (3′FL) and 3′-sialyllactose (3′SL) between centers at tp1, tp3 and tp7; $n = 109$; 3′FL: K1:K3 $q = 1.000$; K1:K7 $q = 1.000$; K3:K7 $q = 1.000$; G1:G3 $q < 0.001$; G1:G7 $q = 0.010$; G3:G7 $q = 1.000$; L1:L3 $q < 0.001$; L1:L7 $q < 0.001$; L3:L7 $q = 1.000$; G1:K1 $q = 1.000$; G3:K3 $q = 0.002$; G7:K7

$q = 1.000$; L1:K1 $q = 1.000$; L3:K3 $q = 0.002$; L7:K7 $q = 1.000$; G1:L1 $q = 1.000$; G3:L3 $q = 1.000$; G7:L7 $q = 1.000$; 3′SL: K1:K3 $q = 0.207$; K1:K7 $q = 0.021$; K3:K7 $q = 1.000$; G1:G3 $q = 1.000$; G1:G7 $q = 1.000$; G3:G7 $q = 0.830$; L1:L3 $q = 1.000$; L1:L7 $q = 1.000$; L3:L7 $q = 1.000$; G1:K1 $q = 1.000$; G3:K3 $q = 1.000$; G7:K7 $q = 0.011$; L1:K1 $q = 1.000$; L3:K3 $q < 0.001$; L7:K7 $q = 0.002$; G1:L1 $q = 1.000$; G3:L3 $q = 0.011$; G7:L7 $q = 1.000$; **d** circle packing plot displaying the numbers of hits for HMO gene clusters found in MAGs (metagenome assembled genomes) and contigs at the different time points. Each circle represents one infant. Colours of the dots indicate where the hit occurred, on MAGs (green) or if it could be only assigned to a contig (yellow) or none (grey); the size of the dots indicate the number of hits; G in red, K in blue, L in green; significance levels are indicated with asterisks for $q < 0.001$ (***), $q < 0.01$ (**), $q < 0.05$ (*) for two-sided t-test, corrected for multiple comparisons with Bonferroni. Centers are abbreviated by G (Graz), L (Leoben), and K (Klagenfurt). For boxplots, the upper, middle, and lower horizontal lines of the box represent the upper, median, and lower quartile; their whiskers depict the smallest or largest values within 1.5-fold of the interquartile range. Source data are provided as a Source Data file (see Github repository).

microbiome composition and function. Of the AMR signatures detected, most were positively correlated with the potential NEC/sepsis pathogens *Enterococcus*, *Staphylococcus* and *Escherichia*, including resistance against β-lactams and erythromycin (Fig. 5b). In contrast,

AMR profiles were highly limited in the probiotic genera *Bifidobacterium* and *Lactobacillus*, underscoring the safety of probiotic administration. *Geobacillus* was also found to not encode AMR genes; therefore, it did not appear to pose an increased health risk.

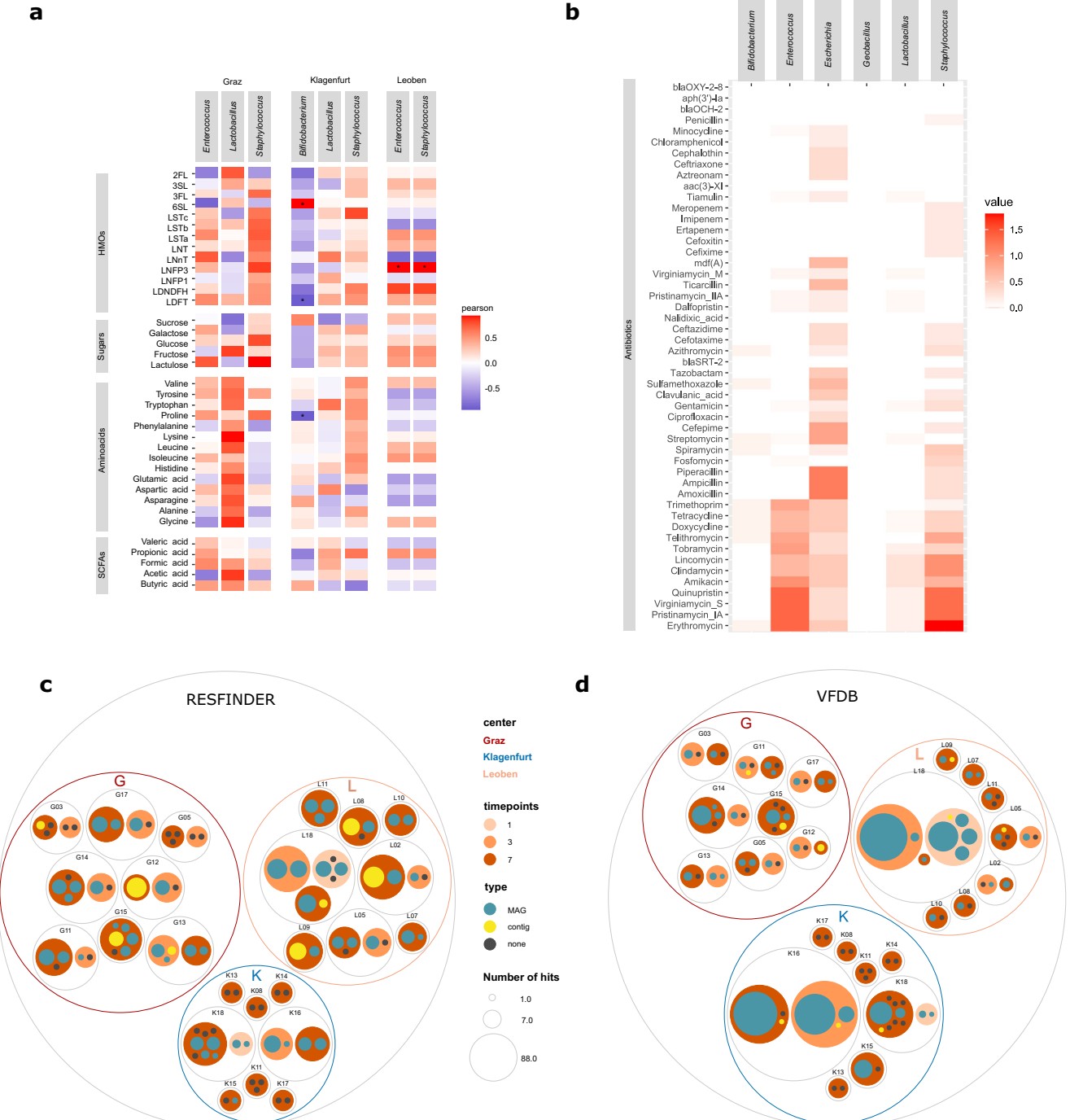

**Fig. 5 | Results of taxonomic correlation analyses and distribution patterns of metabolites, antibiotic resistances and virulence factors. a** Metabolites measured with NMR correlated with bacterial key genera in the three centers; metabolites of the groups of human milk oligosaccharides (HMOs), sugars, amino acids and short-chain fatty acids (SCFAs); **b** correlation of antibiotic resistance genes with bacterial key genera. Circle packing plot of **c** resistance genes and **d** virulence factors in the centers. Each circle represents an infant and is split into the three MGS sequenced time points. Colours of the dots indicate where the hit occurred, on MAGs (metagenome assembled genome) (green) or if it could be only assigned to a contig (yellow) or none (grey); the size of the dots indicates the number of hits. Significance levels are indicated with asterisks for $q < 0.001$ (***), $q < 0.01$ (**), $q < 0.05$ (*) by Pearson corrected for multiple testing by Benjamini Hochberg. Centers are abbreviated by G (Graz), L (Leoben), and K (Klagenfurt). Source data are provided as a Source Data file (see Github repository).

The virulence factor analysis results show that K infants had fewer pathogenic factors in their microbiomes than G and L infants (G: 64 hits, K: 193 hits, L: 173 hits) (Fig. 5d). Enrichment in *E. coli*-associated virulence traits was also observed in two infants in K and L (K16 and L18). Only six out of 431 hits were observed for *Staphylococcus* in six infants. We can assume that *Staphylococcus* is a key player in the context of its abundance, but not in the context of either replication or

virulence factors. As expected, no virulence factors were found for the other key players, including *Bifidobacterium*, *Lactobacillus*, and *Geobacillus*.

## Discussion

NEC prophylaxis and therapy have become a central aspect in the clinical management of VLBW preterm infants. Although probiotics,

human milk, and antibiotics are used by many NICUs to prevent NEC, to our knowledge, an in-depth, systematic comparison of different preventive regimens had not yet been performed. In this study, we show that NEC prophylaxis not only impacts the bacterial gut microbiome composition and function strongly, but also drives strong center-specific patterns observed in the fungal, archaeal and viral parts of the microbiome. This is particularly important with respect to archaea and phages/viruses, as NEC prophylaxes are not directly administered to change this part of the microbiome.

Our iRep analyses and the steady increase in relative abundance (Fig. 2b) suggest that *B. infantis* NCDO 2203 might be actively replicating and thus colonizing the GIT during the administration period. This could be supported by the presence of HMO gene clusters and low levels of HMOs in the stool. Nevertheless, iRep is no sufficient approach to conclude on colonization potential. Although *Bifidobacterium* does not yet appear to naturally colonize the GIT of preterm infants in large numbers, the reliable health benefits of supplemented *Bifidobacterium* have been demonstrated in multiple ways[37], including the negative correlation between *Bifidobacterium* and opportunistic pathogens[38].

Unlike bifidobacteria, a variety of lactobacilli, including *L. acidophilus* NCDO 1748 (administered in center K) and especially *L. rhamnosus* LCR 35 (administered in G), have been observed as natural colonizers of the GIT of preterm infants. In this study, we consider human milk (HM) to be the most likely source, as HM naturally contains large amounts of live lactobacilli[39]. For example, *L. rhamnosus* was successfully isolated from 8.13% of all human milk samples examined by Lubiech et al.[29] and was also detected in culture-independent assays alongside other, more frequently occurring *Lactobacillus* species (*L. salivarius*, *L. fermenting*, *L. gasseri*[17,29,40]). It should be noted, however, that the HM was pasteurized, especially if it came from a milk bank, and pasteurization may obviously reduce the chance that live lactobacilli are transmitted. As all except one infant in the metagenomic subset were born via C-section and not vaginally, vertical transfer from the maternal vaginal microbiome is untenable in this case.

The dynamics of naturally occurring and supplemented (non-natural) probiotic strains are interesting to study, as supplemented bacteria have to "invade" an ecosystem that is evolving to provide colonization resistance against other, potentially pathogenic bacteria. To avoid disrupting this process, and also considering that naturally occurring bacterial residents are likely to persist over a longer period of time[41], an ecologically oriented probiotic choice would tend to include supplementation with *L. rhamnosus* LCR 35 or another beneficial preterm *Lactobacillus* strain.

However, we and others observed that the co-administration of *B. infantis* and *L. acidophilus* in equal amounts results in the overgrowth and predominance of *B. infantis* over *L. acidophilus*[6,42]. The absence of replicating *Lactobacillus* MAGs in the presence of replicating *Bifidobacterium* MAGs leads us to hypothesize that *L. acidophilus* NCDO 1748 cannot successfully colonize the GIT of preterm infants when *B. infantis* NCDO 2203 is co-administered. On the other hand, it may be that *Lactobacillus* plays a pioneering role in anaerobic bifidobacterial colonization by removing oxygen from the GIT[43]. This supporting role of *Lactobacillus* tends to underscore the benefits of taking a multi-species probiotic approach.

Most importantly, administration of *L. rhamnosus* LCR 35 alone had no appreciable effect on the composition or function of the gut microbiome and did not result in a substantial increase in *Lactobacillus* colonization as seen before[44]. Potentially pathogenic bacteria and antibiotic resistance genes were also not substantially reduced when *L. rhamnosus* LCR 35 was administered alone.

We argue that the GIT of VLBW infants per se is not a "normal" natural habitat, and the action of *Lactobacillus* may be too mild or too slow to rapidly support the establishment of a healthy microbiome. In contrast, *B. infantis* NCDO 2203, although not naturally occurring, seems to be a strong, stable, and reliable keystone microbe of the nearly empty niche of the preterm GIT, overgrowing pathogenic threats, and microorganisms carrying antibiotic resistance genes.

Moreover, *Bifidobacterium* together with *Bacteroides* has been described as an effective converter of HMOs[16,45], producing substantial amounts of beneficial SCFAs[46]. The efficient conversion of HMOs and thus SCFA production is an extremely important process for premature infants, which is underscored by the finding that concentrations of HMOs in HM are substantially increased when the infant is born prematurely[29,47]. In fact, the bifidobacteria found in the faeces of K infants exhibited a marked genetic ability to convert HMOs. This could be related to changes in the GIT environment such as lower pH or improved colonization resistance.

This capacity, however, could not be observed in our metabolomic analyses because the K infants were fed with (HMO-lacking) FM. In contrast, in centers L and G, where HMOs were administered in the natural form of HM, the small natural proportion of microbial HMO converters was too low for observable efficient turnover. Consequently, only the simultaneous administration of HMOs and HMO-converters would result in optimal utilization of the health benefits for the infants. This highlights the importance of combining the right probiotic with the right diet.

Furthermore, clinically relevant findings from our study are related to enteral antibiotic administration, as gentamicin was administered in two NICUs, G and L. Indeed, the prophylactic enteral, but not parenteral, administration of antibiotics has been shown to substantially decrease NEC rates[12,48], which is also reflected in the low number of NEC cases observed in center G[14]. To date, no single causative microbial agent of NEC has been described; thus, the antibiotics used, such as gentamicin, must cover a broad spectrum. Our study was not designed to investigate the performance and efficiency of gentamicin in eliminating specific bacteria. However, we did not find consistent negative correlations between gentamicin administration with certain taxa or the occurrence of gentamicin resistances in the microbiome at tp7. Nevertheless, we cannot rule out a negative effect on concomitantly administered probiotic bacteria.

In several studies, intravenous antibiotic administration at a young age has been associated with adverse health outcomes later in life[49–51]. It is also proposed that antibiotic prophylaxis does not reduce NEC incidences but may rather increase the risk for high-risk premature infants of NEC[5]. Nevertheless, in these studies, antibiotics were administered enteral, not intravenously, which needs to be evaluated strictly differently. Still, the prophylactic use of antibiotics must always be weighed against the potential risk. On the one hand, prophylactic administration of antibiotics probably minimizes the outbreak of pathogenic bacteria, especially since infections in premature infants develop alarmingly rapidly, and the success of treatment is time critical. On the other hand, antibiotics could also suppress the growth of beneficial (probiotic) bacteria, which is also underlined by our study results showing that probiotic *Lactobacillus* and *Bifidobacterium* carry few AMR genes and are therefore more susceptible for antibiotics. The long-term effects of antibiotic administration at such an early, vulnerable age are difficult to predict. In general, AMR is a global threat[52] and their horizontal gene transfer to pathogenic bacteria might also be implicated in NEC.

Our study has several strengths and limitations. Overall, due to the extensive analysis performed, the study cohort was kept rather small, and the survey period was limited to the first weeks of life. Unfortunately, the early (meconium) samples could not be used for metagenomic analyses because of their exceptionally low microbial biomass, so we had to focus on tp7 for detailed functional assessments. We could not draw any conclusions about the colonization potential of the probiotically administered strains, as iRep is no sufficient tool to prove replication or colonization. Due to the study design, we cannot discern

which factor (antibiotics, probiotics, nutrition) is driving which result, as more than one of those factors changes between the centers. However, we successfully and comprehensively conducted a multi-center study in which we analysed the longitudinal composition of the microbiome of 55 VLBW preterm infants using amplicon-based and metagenomic sequencing, which also enabled us to elucidate the contribution of archaea, fungi, and phages. A large wealth of taxonomic and functional data were obtained, and analyses revealed the HMO conversion potential and the emergence of antibiotic resistance. In addition, genetically detected functions could be effectively combined with well-found metabolomic analyses.

Our study provides a solid basis for further evaluation and analyses. We found that the combination of feeding HM and administering *B. infantis* NCDO 2203 during the first weeks of life in VLBW infants could be a promising synergistic approach. Overall, all treatment regimens analysed in this study resulted in NEC rates well below the global average, confirming the very successful and strategic management of this devastating disease in our NICUs.

## Methods

### Study design

We conducted a prospective, triple-center cohort pilot study investigating the gut microbiome of preterm infants with a birthweight <1500 g in three Austrian neonatal intensive care units (NICUs). These centers (Klagenfurt, K; Leoben, L; Graz, G) used different regimens for NEC prophylaxis which are summarized in Table 1 and have been described in detail previously[53,21]. A detailed description of the study design is available in[53] and first results have already been published elsewhere[21].

In G, prophylaxis consisted of administration of probiotic *Lactobacillus rhamnosus* LCR 35 twice a day, nystatin, and enteral gentamicin. Probiotic bacterial species were also administered in K, namely *Bifidobacterium longum* subsp. *infantis* NCDO 2203 and *Lactobacillus acidophilus* NCDO 1748 in combination with fluconazole. In center L, no probiotic species, but enteral gentamicin and nystatin were used. Next to medication and probiotic supplementation, the feeding regimen also differed between the centers. In G and L, mainly human milk (HM) was provided. In K, enteral nutrition consisted mainly of formula milk (FM). The feeding history for each infant and time point is shown in Suppl. Fig. 1.

Between October 2015 and March 2017, stool samples were collected from preterm infants at those three centers. Inclusion criteria were birth weight <1500 g and survival in the first three weeks of life. Clinical data on the infants have been published recently[21], with no significant differences between the centers (APGAR, sex, gestational age, gestational weight), except length of hospital stay (G: 72, K: 68.5, L:58; *p* = 0.04). In case of genetic diseases, syndromes or congenital anomalies or meconium ileus, infants were excluded from the study. A total of 55 infants were included in the study (male = 33, female = 22; age 0–3 weeks). The infants' stool samples were collected every other day from meconium for the first two weeks of life, with each infant providing stool samples at seven time points (time points of samples were slightly variable (see Suppl. Table 2) due to the varying availability of fecal samples; average sampling time points were tp1: within day 1-3; tp2: day 4; tp3: day 6; tp4: day 8; tp5: day 10; tp6: day 12; tp7: day 15). A total of 383 samples and 16 negative controls were collected.

The diagnostic criteria for NEC definition were the same in all three centers and followed the AWMF guideline with Bell criteria with modifications of Walsh. NEC incidence rates are 2.2% in K, 2.7% in G, and 4.6% in L[21]. The study is registered wtihin German Registry of Clinical Trials No.: DRKS00009290 and received ethical approval from the local ethic committees (number 27-366 ex14/15) and written informed consent was obtained from the parents of the infants and participants were not compensated.

## Sample processing

**DNA extraction, sequencing, and metabolomics.** Samples were processed as described in detail earlier[21]. In short, genomic DNA was isolated according to manufacturer's instructions using the Magna-Pure LC DNA Isolation Kit III (Roche).

Targeted amplicon sequencing was performed for three different regions: one using universal primers but mainly targeting bacterial V4 16S rRNA gene sequences (515F/R926, 5′GTGY-CAGCMGCCGCGGTAA3′/5′AGCCGYCAATTYMTTTRAGTTT3′), the other aimed for optimized amplification of archaeal 16S rRNA gene sequences in a nested PCR (PCR1 344F/1041R, 5′ACGGGGYGCAG-CAGGCGCGA3′/5′GGCCATGCACCWCCTCTC3′; PCR2 519F/806R, 5′CAGCMGCCGCGGTAA3′/5′GGACTACVSGGGTATCTAAT3′) and the third of the ITS region of fungi (ITS86F/ITS4R, 5′GTGAATCATC-GAATCTTTGAA3′/5′TCCTCCGCTTATTGATATGC3′)[54,55]. See ref. [56] for detailed primer sequences and PCR protocols. PCRs were run in triplicates and pooled subsequently. No human DNA sequence depletion, enrichment of microbial or viral DNA, or mRNA was performed. Library preparation and sequencing of the amplicons were carried out at the Core Facility Molecular Biology of the Center for Medical Research at the Medical University Graz, Austria. Sequencing was performed in paired-end run mode on an Illumina MiSeq with v3 600 chemistry and 300 bp read length[57]. Raw reads are publicly available at the European Nucleotide Archive PRJEB37883. Raw reads were processed using Qiime2 v2019.1 to v2021.2[58]. Briefly, reads that were first quality filtered with DADA2 v2019.1.0 to v2021.2.0[59] were then denoised into Amplicon Sequence Variants (ASVs). The taxonomy was assigned with a Naive-Bayes classifier based on SILVA 132 for bacterial and archaeal signatures[60,61]. Potential contaminant ASVs were removed considering the sequenced negative controls with the R package decontam v3.9 in prevalence mode, isContaminant setting, and threshold 0.5[62], (https://github.com/benjjneb/decontam/). Subsequently negative controls as well as signatures of chloroplasts and mitochondria were removed manually. As no quantification experiments were applied, relative abundance methods were applied.

The genus *Lactobacillus* has recently been taxonomically restructured[63], in which the genus was divided into 25 separate genera. In this study, we continue to refer to the amended nomenclature and dimension of the genus *Lactobacillus*, as this work is a supplement to the previously published amplicon data of this study, and we prefer to be consistent between those two publications. In addition, the taxonomic assignment of the datasets on which all other analyses are was performed using SILVA 132[60] prior to the renaming event. In this publication, we mention only two representatives of the original *Lactobacillus* genus, namely *Lactobacillus rhamnosus* (now: *Lacticaseibacillus rhamnosus*) and *Lactobacillus acidophilus*, the latter remaining unchanged. It shall be noted, that for the important probiotic representatives of the amended *Lactobacillus* genus, the new names also begin with "L" and the abbreviations of the "*L.*" genus may continue to be used[63].

An initial insight into the bacteriome of the analyzed 55 infants, based on 16S rRNA gene amplicons was provided earlier[21]. In this subsequent data analysis, we intensify the analytical assessments and include shotgun metagenomics and functional metabolomics, to substantially deepen the understanding of the development within the first weeks. Further, we include information on the archaeal, fungal, and viral part of the gut microbiome as well as on their function.

We performed shotgun metagenomic sequencing of a subset of infants for three time points (tp1, tp3, tp7). Sequencing libraries were generated with the Nextera XT Library construction kit (Illumina, Eindhoven, the Netherlands) and sequenced on an Illumina HiSeq (Illumina, Eindhoven, the Netherlands; Macrogen, Seoul, South Korea). The raw reads were quality assessed with fastqc v0.11.8[64] and filtered accordingly with trimmomatic v0.38[65] with a minimal length of 50 bp and a Phred quality score of 20 in a sliding window of 5 bp.

Both probiotics used in this study ("Antibiophilus", containing *Lactobacillus rhamnosus* LCR 35, and "Infloran", containing *Bifidobacterium longum* subsp. *infantis* NCDO2203 and *Lactobacillus acidophilus* NCDO1748) are pharmaceuticals according the Austrian regulations, and as such, their quality is strictly regulated and controlled. Therefore, we proceed on the assumption, that the probiotics contain the labeled strain purely and constant over time. However, to confirm the presence of the signatures of the probiotics in the stool of the infants, we compared their genomic information with our amplicon and metagenomic data. Therefore, amplicon sequences of interest were blasted against the respective 16S rRNA genes of the lactobacilli (*L. rhamnosus* LCR35, accession: EU184020; *L. acidophilus* NCDO1748, accession: ATCC4356), indicating a 100% identity for both. As metagenomic MAGs were available for *Bifidobacterium* from Klagenfurt samples, we used genomic information for comparison, showing 99.97% to 100% similarity (FastANI, *B. longum* subsp. *infantis* NCDO2203, accession: ATCC15696). The results are listed in Suppl. Tables 2a and 2b and on our Github repository (https://github.com/CharlotteJNeumann/preterm_shared)[66].

Samples from time points tp1 and tp3 yielded only small amounts of DNA, so that library preparation or post-sequencing quality filtering failed for most samples. Thus, we focus mainly on data from tp7 for analysis and interpretation.

The obtained reads were analyzed both in a genome- as well as gene-centric way. For the gene-centric approach, data were annotated with diamond v0.9.25[67] using blastx search against NCBInr database from 2019-07-19 and analyzed in the open-submission data platform MG-RAST according to the manual using default settings[68], on taxonomic and functional (SEED subsystems[69]) level.

For the genome-centric analysis the reads were co-assembled with Megahit v1.1.3[70] by using the default setting "meta-sensitive" into contigs which were then binned with MaxBin2 v2.2.4[71]. Potential chimers and contaminations of representative dereplicated MAGs were detected with Genome UNClutterer (GUNC v. 1.0.1)[72] using the default diamond GUNC database 2.0.4, its sensitive mode and a detailed output till species level. Genome chimerism was visualized as interactive html plots for each MAG. Finally, all outputs from GUNC were merged with those from checkM v.1.1.0[73] and all models from checkM2 v.0.1.3[74]. The bins were then de-replicated with dRep v2.0.5[75] to generate a list of representative metagenome assembled genomes (MAGs). Taxonomic classification of those MAGs was performed with GTDB-Tk v1.5.1. Replication rates were determined with iRep v1.1[76]. Indices for MAGs with the following default parameters were included: ≥75% completeness; ≤175 fragments/Mbp sequence; ≤2% contamination; ≥5 kbp scaffold length; min cov = 5; min wins = 0.98; min $r^2$ = 0.9; GC correction min $r^2$ = 0.0.

A subset of 111 stool samples for tp1, tp3 and tp7 from all three centers (corresponding to metagenomic sequencing) were analyzed with untargeted NMR (nuclear magnetic resonance spectroscopy) for several metabolites in house as described previously[77]. In short, methanol water was added to the samples, cells were lysed, lyophilized, and mixed with NMR buffer. NMR was performed on an AVANCE™ Neo Bruker Ultrashield Plus 600 MHz spectrometer equipped with a TXI probe head at 310 K and processed as described elsewhere[78].

### Data analysis, statistics and visualization

Multiple analyses were performed in R v4[79] using the Microbiome Explorer package[80] using CSS normalization and DESeq2 for differentially abundance analyses: microbial composition analysis and visualization as stacked bar charts and correlation analysis of abundance of specific bacterial genera with their phages. Differential abundance was plotted in R[79] using the ggplot2 package[81] and asterisks refer to DESEq2 q-values (q-values <0.05 (*), <0.01 (**) and <0.001 (***)) which are available in our Github repository[66]. A pie chart for the *Lactobacillus* genus was created using Krona charts[82]. The BioEnv Biplot for bacterial dissimilarity of the groups was created using the vegan package[83] in R. Significance of differential abundance was calculated in SPSS v27[84] using a Kruskal-Wallis with Bonferroni correction for multiple comparisons and evaluating significance with q-values <0.05 (*), <0.01 (**) and <0.001 (***).

The network was created by using SparCC[85,86] within the SCNIC tool (Sparse Cooccurrence Network Investigation for Compositional data)[87,88] to calculate co-occurences from CSS normalized metagenomic observations. Apart from default settings we used 10 bootstraps to calculate p-values for the SparCC R value, filtered the dataset with activated –sparcc_filter parameter and used the recommended minimum correlation value of 0.35 to determine edges. Calculated correlations and networks were then visualized in Cytoscape v.3.9.1 in an edge-weighted spring embedded network where nodes represent taxa and edges positive and negative co-occurences according to the SparCC R values.

Permanova was used for analyzing biological and technical variations of the data. For that, the amplicon dataset was CSS normalized (RSV level), before feeding into the R-script provided by Lahti, Shetty et al.: https://microbiome.github.io/tutorials/PERMANOVA.html (microbiome::transform: "compositional", permutations=999, method= bray). Input files (all samples, tp3 samples, tp7 samples, and metadata), including script and output, are provided in the Github repository (https://github.com/CharlotteJNeumann/preterm_shared)[66].

### Metabolite correlation with taxonomic information

Metabolites measured by NMR were then correlated with CLR transformed relative abundance of genera of amplicon sequencing in R[79]. As the centers differed greatly in terms of species present, this analysis was performed separately for each center. Therefore, amplicon data for each center were normalized with bestNormalized[89] and then correlated with Pearson. The list of normalizations bestNormalize chose is available in our Github repository[66]. The analysis was plotted in a heatmap using ggplot2[81]. For each center, only the genera with the highest abundance and differentially abundance representing the six key players were selected based on abundance and DESeq2 (p < 0.001).

### Antibiotic resistance genes counts and virulence factors

MAGs and contigs were aligned against several databases in abricate (Seemann T, Abricate, Github https://github.com/tseemann/abricate) with options mincov = 70 and minid = 70, including HMO gene clusters sequences[90], EcOH[91], VFDB[92], and Resfinder[93]. To correlate antibiotic resistance gene patterns with specific genera, the taxonomy of the MAGs was assigned by GTDB-Tk[94]. The number of hits per infant and per time point is shown in a circle packing plot created with rawgraphs[95]. Data were analyzed on genus level whereas features were only depicted when the genus was represented by more than one MAG. Correlation of antibiotic resistance gene patterns with the six key species were visualized in a heatmap with R[79].

All graphs were combined and assimilated in Inkscape v1.1 (URL: https://inkscape.org/en/ RRID:SCR_014479) to obtain a uniform appearance.

### Statistics & reproducibility

We conducted a prospective, triple-center cohort pilot study. As it was a pilot study, sample size was not pre-determined beforehand. Randomization and blinding of the investigators was not foreseen in the chosen study set-up, as all hospitals use different regimens and this protocol was not changed. A full study flow chart is provided in Suppl. Fig. 6. No data were excluded from the analyses. Overall, the study is considered to be only partially reproducible, as the data are dependent on the study cohort, which was only sampled once within this study, and sampling of cohorts in the same time-window cannot be repeated. However, starting from the raw sequencing data, the analysis is fully reproducible and all required data, scripts, and details are provided.

Statistics mostly focus on single time points, mostly tp7; longitudinal statistics was not performed.

## Reporting summary

Further information on research design is available in the Nature Portfolio Reporting Summary linked to this article.

## Data availability

The raw sequencing reads generated in this study have been deposited in the European Nucleotide Archive Database under accession code PRJEB37883. ASV-tables, sequences of MAGs and metabolomic, as well as metabolomic data and used scripts are openly available and shared via Github (https://github.com/CharlotteJNeumann/preterm_shared)[66]. All files used for the figures are listed in the Source Data File, which is also provided at Github ("list_raw_data_figures.xlsx").

Raw NMR data have been deposited in Metabolites under accession code MTBLS6866. Clinically-relevant, anonymized information on each sample (sex, hospital, birth mode, nutrition, medication etc.) is provided in the metadata table located along with the respective ASV tables in the open Github repository. Source data are provided in the open GitHub repository[66]. Source data are also provided with this paper.

## Code availability

R scripts are openly available and shared via Github (https://github.com/CharlotteJNeumann/preterm_shared)[66].

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

## Acknowledgements

We highly appreciate the contributions of Raimund Kraschl, Claudia Kanduth, and Barbara Hopfer. This research was funded in in part by the Austrian Science Fund (FWF) [DOC 31 DP-iDP and P32697, given to CME]. For the purpose of open access, the author has applied a CC BY public copyright licence to any author-accepted manuscript version arising from this submission. TM was supported by Austrian Science Fund (FWF) grants P28854, I3792 and DK-MCD W1226, the Austrian Research Promotion Agency (FFG) grants 864690 and 870454; the Integrative Metabolism Research Center Graz; Austrian Infrastructure Program 2016/2017, the Styrian Government (*Zukunftsfonds*) and BioTechMed-Graz (Flagship project DYNIMO). BR was supported in part by a grant of the parent association "*Kleine Helden –Initiative für Früh- und Neugeborene*", Graz, Austria. LJH is supported by Wellcome Trust Investigator Awards (100974/C/13/Z and 220876/Z/20/Z); the Biotechnology and Biological Sciences Research Council (BBSRC), Institute Strategic Programme Gut Microbes and Health (BB/R012490/1), and its constituent projects BBS/E/F/000PR10353 and BBS/E/F/000PR10356. The authors acknowledge computational resources of the MedBioNode at the Medical University of Graz and the support of the ZMF Galaxy Team: Core Facility Computational Bioanalytics, Medical University of Graz, funded by the Austrian Federal Ministry of Education, Science, and Research *Hochschulraum-Strukturmittel* 2016 grant as part of BioTechMed Graz. The funding body had no influence on the study, collection, analysis, and interpretation of data or on the manuscript content.

## Author contributions

Conceptualization: B.U., B.R., S.K.K., C.M.E. Methodology: C.J.N., S.K.K., A.M., C.K., R.K., M.D., T.M. Formal Analysis: C.J.N., A.M., C.K., R.K., M.D., M.K., T.M. Investigation: C.J.N., C.M.E. Writing—Original draft: C.J.N., C.M.E. Writing—Review & Editing: all authors. Visualization: C.J.N. Supervision: C.M.E., B.R., B.U., L.H. Project Administration: C.J.N., B.U., C.M.E. Funding Acquisition: B.R., B.U., T.M., C.M.E.

## Competing interests

The authors declare no competing interests.

### Inclusion and diversity statement

We worked to ensure sex balance in the selection of participants, as well as authors. We further actively worked to promote gender balance in our reference list.

## Additional information

[1]Diagnostic and Research Institute of Hygiene, Microbiology and Environmental Medicine; Medical University of Graz, Graz, Styria 8010, Austria. [2]Quadram Institute Bioscience, Norwich Research Park, Norwich NR4 7UQ, UK. [3]Chair of Intestinal Microbiome, School of Life Sciences, ZIEL-Institute for Food & Health; Technical University of Munich, Freising, Bavaria 85354, Germany. [4]Gottfried Schatz Research Center for Cell Signaling, Metabolism and Aging, Molecular Biology & Biochemistry, Medical University of Graz, Graz, Styria 8010, Austria. [5]BioTechMed, Graz, Styria 8010, Austria. [6]Division of Paediatric Cardiology, Department of Paediatrics and Adolescent Medicine, Medical University of Graz, Graz, Styria 8036, Austria. [7]Division of Neonatology; Department of Paediatrics and Adolescent Medicine, Medical University of Graz, Graz, Styria 8036, Austria. [8]Research Unit for Neonatal Infectious Diseases and Epidemiology, Medical University of Graz, Graz, Styria 8036, Austria. [9]Norwich Medical School, University of East Anglia, Norwich Research Park, Norwich NR4 7TJ, UK. ✉e-mail: bernhard.resch@medunigraz.at; christine.moissl-eichinger@medunigraz.at

