## [Peer Review File · Nature Communications]

REVIEWER COMMENTS

Reviewer #1 (Remarks to the Author):

NEC is a severe, life-threatening condition that disproportionately affects low birthweight preterm infants. The authors investigate the impact of 3 preventive regimens on 54 low birthweight preterm infants, on a wide range of aspects: bacterial (via 16S and WGS), fungal, viral and archaeal taxonomic profiling. The authors also investigate virulence factors, antibiotic resistance profiles, HMOs and SCFAs.

Significant variability was found across the 3 regimens and across all kingdoms investigated.

The regimens are not limited to inclusion/exclusion of a specific probiotic supplementation, but include a combination of antibacterial and antifungal treatments, different feeding modes and different administration route. The impossibility to discern what is the effect of the probiotic supplementation vs other interventions limits the biological insight and the portability of the conclusions of this study. This study aims at “serv(ing) as a model for implementing similar strategies to combat NEC in other NICUs worldwide” but the presented results are potentially reproducible only if the exact regimen is followed in other NICUs.

Major concerns:

The study design, as shown in Table 1, presents several significant weaknesses:

- Each group is a combination of different geographical location, diet, probiotic(s), antibiotic and antifungal treatment. In addition, also the administration route of the antifungal treatment differs among groups. In this context, it is impossible to assess the impact of each factor (either alone or in combination) in the presented results.
- No control was included to verify that geographic location was not the main driver of the presented results.
- Considering the high number of variables (listed above), the study is underpowered.
- The probiotics themselves have not been sequenced. Several studies point at the potential inaccuracy of probiotics labelling (missing or different strains, potential contamination and different dosage from what indicated on the label) [1, 2]. This point alone raises questions on the validity of the downstream results.

Despite the extensive multi-kingdom profiling, the manuscript is mostly descriptive, with limited insight into the community dynamics behind the presented results. This is also due to the impossibility to discern which factor is driving which result. An example of this limitation involves one of the two results included in the abstract:

Lines 40-42: “Engraftment of Bifidobacterium substantially reduced microbiome-associated antibiotic resistance as compared to regimens using probiotic Lactobacillus rhamnosus or no supplementation.” As shown in Table 1, Bifidobacterium supplementation is provided in Group K, which is also the only group in which no antibiotics are used. Therefore, is the reduction in antibiotic resistance due to Bifidos colonisation or is it due to the lack of prophylactic antibiotic therapy? The study design does not allow to answer this question properly.

Another example is provided by Fig1B-G and Fig2B, showing that the archaeal, fungal, viral and bacterial profiling of group K (probiotics+, abx-) is significantly different from that of group L (probiotics-, abx+) and group G (probiotics+, abx+). It is not possible to establish if this difference is due to the probiotic itself or the lack of antibiotic treatment.

Minor concerns:

- Due to the significant heterogeneity of probiotic strain functions, mentions in the text should always refer to the exact species and strain investigated. Throughout the text the probiotic supplementation is referred as generic “Lactobacillus ” and “Bifidobacterium”.
- The resolution in all figures is too low to be easily readable.
- Line 141: the authors refer to “previous reports” without including any reference.
- Lines 141-142: The authors present only one side of the narrative (archaea not being present during the first 3yrs of infancy), but some studies found archaea in the gut of infants much earlier than that (as early as the first day of life in [3]). Presenting both aspects helps the reader putting the authors’ finding in the broader context of literature
- Figure 2D: authors should clearly state what is the % threshold used to include a Lactobacillus species in the plot (0.7%?)
- The color palette used throughout the paper is not colorblind friendly (red/green). ColorBrewer offers colorblind safe quantitative palettes [4].
- Bits of text throughout the manuscript are underlined
- The authors refer to all 3 regimens as successful. However, no NEC baseline rate is provided for geographically-matched NICUs not performing any of the 3 regimens evaluated in this study.
- Line 522: Bifidobacterium is not italicized

- MAGs QC based on CheckM is not sufficient to detect chimerism. Please check out [5,6].

References:

- 1:<https://www.ncbi.nlm.nih.gov/pmc/articles/PMC8329331/>
- 2:<https://www.frontiersin.org/articles/10.3389/fmicb.2021.693973/full>
- 3:<https://www.frontiersin.org/articles/10.3389/fmicb.2017.00738/full>
- 4:<https://colorbrewer2.org/#type=qualitative&scheme=Dark2&n=3>
- 5: <https://genomebiology.biomedcentral.com/articles/10.1186/s13059-021-02393-0>
- 6:<https://pubmed.ncbi.nlm.nih.gov/32188701/>

Reviewer #2 (Remarks to the Author):

The manuscript by Neumann et al reports on preterm infant gut microbiome at different time points from three clinical sites in Austria. The manuscript is largely descriptive and builds on previous work by the authors, with the main novelty coming from the use of metagenomics and metabolomics on a subset of the samples. Overall the manuscript is well-written, interesting, and would add value to existing literature. I have some comments below.

Note, as there were no line or page numbers I am unable to easily point to specific elements.

Major comments

The title could be misleading. For instance, suggesting probiotics are a “Successful clinical practices to prevent necrotizing enterocolitis” is not well supported at present. The title would be more appropriate by removing necrotizing enterocolitis since this was not well explored in the current study, and instead wording along the lines of “Different clinical centers and regimens are associated with altered preterm infant gut microbiome”

The study is based on different regimens as outlined in Table 1. The authors should include a cohort table of infants from this specific study and determine if there was a significant difference in e.g., feeding between the centers. This is important as, for instance, the authors claim findings are related to differences in feed, but if the amount of human milk received is comparable across centers then it might not support their narrative. Unless the infants in K never received any human milk, which would seem unlikely and unjustified? In which case this could be clearer.

The authors used either two-sided Welch's t-test or ANOVA for determining significance of taxa. How did they account for multiple comparisons?

Additionally, from what I can tell, the authors included repeated measures in some analyses (e.g., comparing between sites including multiple time points). How did they account for repeated measures?

A challenge of having multiple related datasets is making it clear what dataset is being referred to within the text. I found this difficult at times, for instance when the authors describe changes in taxa (e.g., in the section "Supplemented Bifidobacterium suppresses natural pathobiont colonizers and co-administered lactobacilli") are they referring to 16S or metagenomics, or both interchangeably?

The authors claim probiotic *B. infantis* suppressed *L. acidophilus*. However, this is based on relative abundance data and has some assumptions. For instance, did the authors culture the probiotic to show both species were present in equal amounts viably? If not, perhaps *B. infantis* is actually in higher numbers in the probiotic. The term 'suppression' also suggests *B. infantis* is actively suppressing *L. acidophilus*, but since this is relative abundance data it more likely reflects that the preterm gut ecosystem and availability nutrients (e.g., HMOs) is better suited to support the growth of *B. infantis*. That numerous publications show *Lactobacillus* do not typically colonise the preterm gut in large numbers further supports this. Although this active 'suppression' could be further tested experimentally in competitive co-culture assays.

Contamination of probiotic products has been reported. Did the authors also sequence the probiotic products?

The sentence "Lactobacillus rhamnosus ... suggesting that this species is indeed part of the natural infant gut microbiome of preterm infants and is likely transmitted through breastfeeding" is not supported in the current data and the reference links to a review that does not report on preterm

infants. Preterm milk microbiome studies are few, but there is not strong evidence that *Lactobacillus* is detectable in preterm milk. Thus, the authors cannot make this claim in the results section without sequencing milk samples. Furthermore, they note in the discussion the milk may have undergone pasteurization which would again remove all viable *Lactobacillus*.

The authors use iRep to indicate “successful niche colonization”, but this alone does not show colonization per se. The authors would need to determine if the probiotic species are detected after probiotics are stopped.

The section “Low level occurrence of diverse types of potentially pathogenic bacteria” is based on the authors selecting specific taxa previously associated with NEC. However, there are also reports of some of these taxa being associated with health. In addition, only a single baby with NEC was included in the analysis. Thus, this section detracts from the overall reporting and should be removed or significantly expanded to include more NEC cases.

Figure 3B reports *Salmonella* and multiple *Shigella* species. Are the preterm infants in this cohort really colonized by these species?

The authors report “K samples showed an overall significantly reduced level of genes involved in oxidative stress, osmotic stress, acid stress and respiration (Welch’s t-test, $p < 0.01$), indicating early maturation of the infant gut and successful establishment of anaerobic conditions.” First, the authors need to expand on how lower oxidative stress and other genes in the microbiome link to early maturation? What is the evidence for this? And equally how does this show successful establishment of anaerobic conditions?

They note “Our findings show that *Bifidobacterium* supplementation (as compared with other probiotics, nutritional and antibiotic treatments) appears to have the most beneficial effects on the early gastrointestinal microbiome, as reflected by centre K having the lowest NEC rates”. This is strong statement and one wonders if the NEC rates significantly reduced after the probiotics were introduced into K. If not, it is misleading to relate the lower NEC rates specifically to the use of probiotics.

They conclude “treatment regimens analysed in this study resulted in NEC rates well below the global average, confirming the excellent and strategic management of this devastating disease in our NICUs. These therapies, or perhaps a novel combination thereof, could serve as a model for implementing similar strategies to combat NEC in other NICUs worldwide.” Many NICUs do use similar regimens and so to suggest that this factor alone will improve NEC rates is oversimplified and

misleading, and also ignores bodies of evidence showing no impacts of probiotics and also the potential for probiotic sepsis.

Minor comments

Typo “ssp” in abstract should be “subsp”

Typo in methods “Shortly” should presumably be “briefly” or “In short”

The authors reference other papers for methods but all the information should be contained in the current manuscript so the reader can easily access this information.

The introduction is comprehensive but very long and I would suggest condensing substantially.

The authors use phrases such “global effect”, however, their data is “association” and unable to determine cause or effect. The authors should ensure they do not suggest causation without further work and using terms such as associated would be more appropriate.

“following chapter” – This wording is incorrect since there are no chapters. Do the authors mean next “section”?

When the authors introduce they did NMR they need to specify how many samples and which time points.

Rebuttal letter

We appreciate the time and effort of both reviewers and are thankful for the helpful comments and corrections. Please note that the provided Line numbers refer to the version with track-changes.

Reviewer 1

NEC is a severe, life-threatening condition that disproportionately affects low birthweight preterm infants. The authors investigate the impact of 3 preventive regimens on 54 low birthweight preterm infants, on a wide range of aspects: bacterial (via 16S and WGS), fungal, viral and archaeal taxonomic profiling. The authors also investigate virulence factors, antibiotic resistance profiles, HMOs and SCFAs.

Significant variability was found across the 3 regimens and across all kingdoms investigated. The regimens are not limited to inclusion/exclusion of a specific probiotic supplementation, but include a combination of antibacterial and antifungal treatments, different feeding modes and different administration route. The impossibility to discern what is the effect of the probiotic supplementation vs other interventions limits the biological insight and the portability of the conclusions of this study. This study aims at “serv(ing) as a model for implementing similar strategies to combat NEC in other NICUs worldwide” but the presented results are potentially reproducible only if the exact regimen is followed in other NICUs.

Response: we appreciate the reviewer’s comments. Please find our point-by-point responses below.

Major concerns:

The study design, as shown in Table 1, presents several significant weaknesses:

- Each group is a combination of different geographical location, diet, probiotic(s), antibiotic and antifungal treatment. In addition, also the administration route of the antifungal treatment differs among groups. In this context, it is impossible to assess the impact of each factor (either alone or in combination) in the presented results.

Our aim was to analyze the situation of applied clinical practices that have been used for a long time in the observed neonatal intensive care units. We think that the focus on clinical practices as they are applied is a particular strength of our study.

We agree that an appropriate scientific approach would have been to modulate single aspects, preferably in the same hospital setting. However, recent publications have discussed cross-contamination issues when probiotics studies are conducted in the same hospital (see L109-111). Our clinical coauthors also address the ethical aspect: it would be unethical to create new artificial combinations of individual factors that might not be expected to have beneficial effects on infants.

In view of both aspects, we decided to analyze the situation as it is, but to select clinical centers wisely and in close geographic proximity (see further details on this raised concern below).

However, we agree that, because of the different circumstances in each clinical center, no clear conclusions can be drawn about the influence of individual factors. In response to your concerns, we have tempered our conclusions accordingly throughout the manuscript (e.g., L 38, 463), and consider the variables to be part of a larger regimen (set of variables) in each clinic.

We further included an additional sentence at the beginning of the Results section: L 124 to 129.: “It should be mentioned that the three clinical situations studied here differed in a number of confounding factors, both recorded and possibly unrecorded, and we can only draw conclusions based on the overall setting in each NICU, which includes medication and probiotic regimens.”

We hope that our changes address the reviewer's concern.

- No control was included to verify that geographic location was not the main driver of the presented results.

We appreciate your response and are grateful for the opportunity to clarify this potential issue. The three neonatal intensive care units are geographically very close to each other and are no more than 100 km apart. The catchment area of the patients also overlaps, so the origin of the patients does not necessarily correspond to the geography of the hospitals. In general, the south of Austria (especially in this area) is very homogeneous in terms of environment and sociodemographic patterns, so we did not consider geography as the main cause for our microbiome differences.

We included an additional sentence at the beginning of the Results section: L127 to 129: “However, the three hospitals are geographically very close to each other, so the patient catchment area also overlaps, and we may consider other factors to be minor compared with the medication and feeding protocols.”

- Considering the high number of variables (listed above), the study is underpowered.

In total, we analyzed 383 samples from patients across the three centers, giving us sufficient statistical power to observe robust, statistically significant differences between the groups. We have tempered our conclusions regarding the contribution of individual variables (see commentary above) so that we rather focus more on the influence of the regimens of the three NICUs as a whole.

- The probiotics themselves have not been sequenced. Several studies point at the potential inaccuracy of probiotics labelling (missing or different strains, potential contamination and different dosage from what indicated on the label) [1, 2]. This point alone raises questions on the validity of the downstream results.

Our two probiotic products are approved pharmaceuticals (products: Infloran and Antibiohilus) and not dietary supplements (as used in most probiotic studies). As medicinal products, the two probiotics are subject to strict regulations to ensure the stability and quality of the product: the content (to detect possible impurities) is monitored through frequent characterization, and both, stability and purity are strictly controlled. More information on the Austrian regulations for “Arzneimittel” and their quality management can be found here:

<https://www.ris.bka.gv.at/GeltendeFassung.wxe?Abfrage=Bundesnormen&Gesetzesnummer=10010441>

However, to address the reviewer's concern, we added additional analyses to confirm that the observed sequences of corresponding bifidobacteria and lactobacilli signatures are indeed stemming from the administered probiotic taxa. These sentences were added to the Materials and Methods section:

“Both probiotics used in this study (“Antibiophilus”, containing *Lactobacillus rhamnosus* LCR 35, and “Infloran”, containing *Bifidobacterium longum* subsp. *infantis* NCDO2203 and *Lactobacillus acidophilus* NCDO1748) are pharmaceuticals according the Austrian regulations, and as such, their quality is strictly regulated and controlled. Therefore, we proceed on the assumption, that the probiotics contain the labeled strain purely and constant over time. However, to confirm the presence of the signatures of the probiotics in the stool of the infants, we compared their genomic information with our amplicon and metagenomic data. Therefore, amplicon sequences of interest were blasted against the respective 16S rRNA genes of the lactobacilli (*L. rhamnosus* LCR35, accession: EU184020; *L. acidophilus* NCDO1748, accession: ATCC4356), indicating a 100% identity for both. As MAGs were available for *Bifidobacterium* from Klagenfurt samples, we used full genomic information for comparison, showing 99.97 to 100% similarity (FastANI, *B. longum* subsp. *infantis* NCDO2203, accession: ATCC15696). The results are listed in SupplementaryTables 2a and 2b and on github. “

- Despite the extensive multi-kingdom profiling, the manuscript is mostly descriptive, with limited insight into the community dynamics behind the presented results. This is also due to the impossibility to discern which factor is driving which result. An example of this limitation involves one of the two results included in the abstract:

Lines 40-42: “Engraftment of *Bifidobacterium* substantially reduced microbiome-associated antibiotic resistance as compared to regimens using probiotic *Lactobacillus rhamnosus* or no supplementation.” As shown in Table 1, *Bifidobacterium* supplementation is provided in Group K, which is also the only group in which no antibiotics are used. Therefore, is the reduction in antibiotic resistance due to *Bifidos* colonisation or is it due to the lack of prophylactic antibiotic therapy? The study design does not allow to answer this question properly.

Another example is provided by Fig1B-G and Fig2B, showing that the archaeal, fungal, viral and bacterial profiling of group K (probiotics+, abx-) is significantly different from that of group L (probiotics-, abx+) and group G (probiotics+, abx+). It is not possible to establish if this difference is due to the probiotic itself or the lack of antibiotic treatment.

Thank you for this comment, and we agree. We toned down our statements accordingly. Additionally, we added the fact, that we cannot discern which factor is driving which result, to the limitation section: L 558-560 “Due to the study design, we cannot discern which factor (antibiotics, probiotics, nutrition) is driving which result, as more than one of those factors changes between the centers.”

As well, we added two sentences about this to the beginning of the results section. L 124-129:

“It should be mentioned that the three clinical situations studied here differed in a number of confounding factors, both recorded and possibly unrecorded, and we can only draw conclusions based on the overall setting in each NICU, which includes medication and probiotic regimens. However, the three hospitals are geographically very close to each other, so the patient catchment area also overlaps, and we may consider other factors to be minor compared with the medication and feeding protocols.”

Minor concerns:

- Due to the significant heterogeneity of probiotic strain functions, mentions in the text should always refer to the exact species and strain investigated. Throughout the text the probiotic supplementation is referred as generic “*Lactobacillus*” and “*Bifidobacterium*”.

We corrected that by including species and strain specification.

- The resolution in all figures is too low to be easily readable.

We are sorry, that the resolution in the file you received is too low to be easily readable. The submitted files have the required dpi 300 and we hope that this is sufficient. If this is not a problem of the file format you received, but of the figures themselves, we are happy to adjust them accordingly.

- Line 141: the authors refer to “previous reports” without including any reference.

We included the reference.

- Lines 141-142: The authors present only one side of the narrative (archaea not being present during the first 3yrs of infancy), but some studies found archaea in the gut of infants much earlier than that (as early as the first day of life in [3]). Presenting both aspects helps the reader putting the authors’ finding in the broader context of literature

We included a brief context with providing literature presenting both sides.

- Figure 2D: authors should clearly state what is the % threshold used to include a Lactobacillus species in the plot (0.7%?)

The threshold was not chosen on purpose, but selected by the KronaChart software to make the figure appear more convenient. In case of more detailed interest, the list of all Lactobacillus species is available in our taxonomic table on github (https://github.com/CharlotteJNeumann/preterm_shared/tree/main/Metagenomics/taxonomy)

- The color palette used throughout the paper is not colorblind friendly (red/green). ColorBrewer offers colorblind safe quantitative palettes [4].

We adjusted the colors accordingly.

- Bits of text throughout the manuscript are underlined

We removed the underlined format.

- The authors refer to all 3 regimens as successful. However, no NEC baseline rate is provided for geographically-matched NICUs not performing any of the 3 regimens evaluated in this study.

We referred to as “successful” in the context of global NEC rates. As this is indeed somewhat misleading, we changed the title accordingly.

Within the region analyzed, unfortunately no additional NICUs are available to perform geographical matching with clinics that are not using any intervention protocols. However, Alsaied et al., (<https://bmcpediatr.biomedcentral.com/articles/10.1186/s12887-020-02231-5>), clearly state, that NEC subgroup analysis based on geographic regions did not reveal any differences, indicating that the geography itself is not a relevant confounding factor. The authors conclude that major differences are due to clinical and health settings in addition to methodological variations.

- Line 522: Bifidobacterium is not italicized

We corrected that.

- MAGs QC based on CheckM is not sufficient to detect chimerism. Please check out [5,6].

We included GUNC and CheckM2 accordingly to detect chimerism. The data can be found in the Supplementary Information and the Material and Methods part was extended accordingly.

References:

- 1: <https://www.ncbi.nlm.nih.gov/pmc/articles/PMC8329331/>
- 2: <https://www.frontiersin.org/articles/10.3389/fmicb.2021.693973/full>
- 3: <https://www.frontiersin.org/articles/10.3389/fmicb.2017.00738/full>
- 4: <https://colorbrewer2.org/#type=qualitative&scheme=Dark2&n=3>
- 5: <https://genomebiology.biomedcentral.com/articles/10.1186/s13059-021-02393-0>
- 6: <https://pubmed.ncbi.nlm.nih.gov/32188701/>

Reviewer 2

The manuscript by Neumann et al reports on preterm infant gut microbiome at different time points from three clinical sites in Austria. The manuscript is largely descriptive and builds on previous work by the authors, with the main novelty coming from the use of metagenomics and metabolomics on a subset of the samples. Overall the manuscript is well-written, interesting, and would add value to existing literature. I have some comments below.

Note, as there were no line or page numbers I am unable to easily point to specific elements.

We apologize for the inconvenience; we have now included line numbers. We highly appreciate your response and comments, which are all addressed point-by-point below.

Major comments

- The title could be misleading. For instance, suggesting probiotics are a “Successful clinical practices to prevent necrotizing enterocolitis” is not well supported at present. The title would be more appropriate by removing necrotizing enterocolitis since this was not well explored in the current study, and instead wording along the lines of “Different clinical centers and regimens are associated with altered preterm infant gut microbiome”

We are grateful for pointing us to this problematic issue.

The phrase “successful clinical practices to prevent necrotizing enterocolitis” refers to the fact that we are comparing NEC prevention practices that are driven in the NICUs to prevent NEC and that are successful in a global comparison. We changed the title accordingly to “Clinical necrotizing enterocolitis prevention practices drive different microbiome profiles and functional responses in the preterm intestine” to make the aim and result of our study clearer: not to show that and how successful the practices are, but that those practices drive different microbial profiles and functional responses.

- The study is based on different regimens as outlined in Table 1. The authors should include a cohort table of infants from this specific study and determine if there was a significant difference in e.g., feeding between the centers. Refer to study protocol or include table again. This is important as, for instance, the authors claim findings are related to differences in feed, but if the amount of human milk received is comparable across centers then it might not support their narrative. Unless the infants in K never received any human milk, which would seem unlikely and unjustified? In which case this could be clearer.

We inserted a reference to table 2 in the study protocol, which shows that there are no statistically significant differences between the study groups in the observed metadata, except for length of hospital stay of the mothers. L 122-123:

“The study groups do not differ statistically significantly in any observed metadata except for the length of hospital stay of the mothers after birth ⁽²¹⁾, Table 2 therein.”

*A figure of feeding history per infant and time point was added in the Supplementary Information and mentioned in the material and method part “The feeding history for each infant and timepoint is shown in **Suppl. Fig. 1**” (L681)*

- The authors used either two-sided Welch's t-test or ANOVA for determining significance of taxa. How did they account for multiple comparisons?

Bonferroni-correction was applied to correct for multiple comparisons. The information was added to the manuscript and the q-values were adjusted accordingly.

- Additionally, from what I can tell, the authors included repeated measures in some analyses (e.g., comparing between sites including multiple time points). How did they account for repeated measures?

In the case of the HMO metabolites (Fig. 4B), we actually included multiple time points per site. We decided to modify this figure by including only tp7, as we did in all other analyses, to maintain a consistent approach. Therefore, no correction for repeated measurements is required.

- A challenge of having multiple related datasets is making it clear what dataset is being referred to within the text. I found this difficult at times, for instance when the authors describe changes in taxa (e.g., in the section “Supplemented Bifidobacterium suppresses natural pathobiont colonizers and co-administered lactobacilli”) are they referring to 16S or metagenomics, or both interchangeably?

We have added the appropriate information in the figure legends and in the body of the manuscript wherever necessary.

- The authors claim probiotic *B. infantis* suppressed *L. acidophilus*. However, this is based on relative abundance data and has some assumptions. For instance, did the authors culture the probiotic to show both species were present in equal amounts viably? If not, perhaps *B. infantis* is actually in higher numbers in the probiotic. The term ‘suppression’ also suggests *B. infantis* is actively suppressing *L. acidophilus*, but since this is relative abundance data it more likely reflects that the preterm gut ecosystem and availability nutrients (e.g., HMOs) is better suited to support the growth of *B. infantis*. That numerous publications show *Lactobacillus* do not typically colonise the preterm gut in large numbers further supports this. Although this active ‘suppression’ could be further tested experimentally in competitive co-culture assays.

We agree with the reviewer, that the term “suppress” was not wisely chosen, and implies active suppression, although this was not actively tested. We toned down our statements respectively (e.g., L 197).

We did not cultivate the probiotics, as the probiotics used are Arzneimittel following the Austrian law (please see the comments to Reviewer 1 for more detail, and below), and thus underlie strict quality control.

- Contamination of probiotic products has been reported. Did the authors also sequence the probiotic products?

Our two probiotic products are approved pharmaceuticals (products: Infloran and Antibiohilus) and not dietary supplements (as used in most probiotic studies). As medicinal products, the two probiotics are subject to strict regulations to ensure the stability and quality of the product: the content (to detect possible impurities) is monitored through frequent characterization, and both, stability and purity are strictly controlled. More information on the Austrian regulations for “Arzneimittel” and their quality management can be found here:

<https://www.ris.bka.gv.at/GeltendeFassung.wxe?Abfrage=Bundesnormen&Gesetzesnummer=10010441>

However, to address the reviewer's concern, we added additional analyses to confirm that the observed sequences of corresponding bifidobacteria and lactobacilli signatures are indeed stemming from the administered probiotic taxa. These sentences were added to the Materials and Methods section:

*"Both probiotics used in this study ("Antibiophilus", containing *Lactobacillus rhamnosus* LCR 35, and "Infloran", containing *Bifidobacterium longum* subsp. *infantis* NCDO2203 and *Lactobacillus acidophilus* NCDO1748) are pharmaceuticals according to the Austrian regulations, and as such, their quality is strictly regulated and controlled. Therefore, we proceed on the assumption, that the probiotics contain the labeled strain purely and constant over time. However, to confirm the presence of the signatures of the probiotics in the stool of the infants, we compared their genomic information with our amplicon and metagenomic data. Therefore, amplicon sequences of interest were blasted against the respective 16S rRNA genes of the lactobacilli (*L. rhamnosus* LCR35, accession: EU184020; *L. acidophilus* NCDO1748, accession: ATCC4356), indicating a 100% identity for both. As MAGs were available for *Bifidobacterium* from Klagenfurt samples, we used full genomic information for comparison, showing 99.97 to 100% similarity (FastANI, *B. longum* subsp. *infantis* NCDO2203, accession: ATCC15696). The results are listed in Supplementary Tables 2a and 2b and on github. "*

- The sentence "Lactobacillus rhamnosus ... suggesting that this species is indeed part of the natural infant gut microbiome of preterm infants and is likely transmitted through breastfeeding" is not supported in the current data and the reference links to a review that does not report on preterm infants. Preterm milk microbiome studies are few, but there is not strong evidence that Lactobacillus is detectable in preterm milk. Thus, the authors cannot make this claim in the results section without sequencing milk samples. Furthermore, they note in the discussion the milk may have undergone pasteurization which would again remove all viable Lactobacillus.

The new feeding-history figure shows, that indeed no pasteurized HM was fed in Leoben. The statement was corrected/adapted accordingly, by including "transmitted through breastfeeding or other sources".

- The authors use iRep to indicate "successful niche colonization", but this alone does not show colonization per se. The authors would need to determine if the probiotic species are detected after probiotics are stopped.

We toned this section down in our discussion (II467 - 469) and added a statement to the limitation section (II 560-564).

- The section "Low level occurrence of diverse types of potentially pathogenic bacteria" is based on the authors selecting specific taxa previously associated with NEC. However, there are also reports of some of these taxa being associated with health. In addition, only a single baby with NEC was included in the analysis. Thus, this section detracts from the overall reporting and should be removed or significantly expanded to include more NEC cases.

We think that the readers might be interested in seeing the effect of different NEC prevention strategies on typical NEC taxa. The selection of those specific taxa that were previously associated with NEC was based on a review that we consider a trustworthy reference.

We believe that the brief chapter on NEC taxa is a good way to incorporate the results of a previous comprehensive review into our data analysis and to present the presence of specific taxa that have been associated with NEC in published research by other research groups. We do not claim that these taxa cause NEC, nor do we perform any statistical analysis or inference on them, because we indeed have only one NEC case in the data set. The goal of this analysis was not to diagnose NEC, which would not be possible given the NEC numbers, but rather to compare patterns between centers.

Reviewer 1 did not comment negatively on this analysis, and we think it still provides valuable insight. We suggest that it be left to the editor to decide whether or not the analysis should remain in the manuscript. We welcome further advice.

- Figure 3B reports Salmonella and multiple Shigella species. Are the preterm infants in this cohort really colonized by these species?

Based on your bioinformatic analysis, the signatures found were assigned to these species, which does, of course, not infer colonization of the infants. In general, the classification of Escherichia/Shigella is, due to their genetic relationship, highly difficult and would require additional analyses, which we did not perform. We added a comment accordingly, however we dare to change the classification received from the bioinformatic pipeline without further analyses. L 289:

“We note that the classification of the microbial species is based on the automated output during our microbiome analyses (see Material and Methods) and does not mean that the children were indeed colonized by potentially pathogenic species.”

- The authors report “K samples showed an overall significantly reduced level of genes involved in oxidative stress, osmotic stress, acid stress and respiration (Welch’s t-test, $p < 0.01$), indicating early maturation of the infant gut and successful establishment of anaerobic conditions.” First, the authors need to expand on how lower oxidative stress and other genes in the microbiome link to early maturation? What is the evidence for this? And equally how does this show successful establishment of anaerobic conditions?

We expanded our chain of thoughts and accordingly, toned down the conclusion.

- They note “Our findings show that Bifidobacterium supplementation (as compared with other probiotics, nutritional and antibiotic treatments) appears to have the most beneficial effects on the early gastrointestinal microbiome, as reflected by centre K having the lowest NEC rates”. This is strong statement and one wonders if the NEC rates significantly reduced after the probiotics were introduced into K. If not, it is misleading to relate the lower NEC rates specifically to the use of probiotics.

The statement was changed to: “Our findings show that regimens including B. infantis NCD0 2203 supplementation (as compared with other probiotics, nutritional and antibiotic treatments) appear to have a highly beneficial effect on the early gastrointestinal microbiome, as reflected by centre K having the lowest NEC rates.” (L465)

The relation is now made on the overall regimen, not the probiotics itself.

- They conclude “treatment regimens analysed in this study resulted in NEC rates well below the global average, confirming the excellent and strategic management of this devastating disease in our NICUs. These therapies, or perhaps a novel combination thereof, could serve as a model for

implementing similar strategies to combat NEC in other NICUs worldwide.” Many NICUs do use similar regimens and so to suggest that this factor alone will improve NEC rates is oversimplified and misleading, and also ignores bodies of evidence showing no impacts of probiotics and also the potential for probiotic sepsis.

We agree, and we toned down this section and removed the last sentence. L 571 – 577.

Minor comments

- Typo “ssp” in abstract should be “subsp”

This was corrected throughout.

- Typo in methods “Shortly” should presumably be “briefly” or “In short”

This was corrected.

- The authors reference other papers for methods but all the information should be contained in the current manuscript so the reader can easily access this information.

We extended the methods section accordingly.

- The introduction is comprehensive but very long and I would suggest condensing substantially.

We shortened the introduction wherever possible.

- The authors use phrases such “global effect”, however, their data is “association” and unable to determine cause or effect. The authors should ensure they do not suggest causation without further work and using terms such as associated would be more appropriate.

We toned down our conclusions accordingly (L 133).

- “following chapter” – This wording is incorrect since there are no chapters. Do the authors mean next “section”?

This was corrected.

- When the authors introduce they did NMR they need to specify how many samples and which time points.

This information was already included in the Material and Method section, but also included it now into the result section when introducing NMR (L.353)

Reviewers' Comments:

Reviewer #1:

Remarks to the Author:

Overall, the authors have addressed most of the comments. Few additional minor comments below:

- The quality of the figures included in the pdf document with the main text has actually worsened compared to the original submission. The authors say they have submitted the files in the required 300dpi format, so the issue might only be in the inclusion of the figures in the main text document. Please make sure to fix this before publication.

- "suppl. table 2" referred to in the main text does not exist anymore. Please include where appropriate the reference to suppl. table 2a and 2b.

- The authors replied: "We included GUNC and CheckM2 accordingly to detect chimerism. The data can be found in the

Supplementary Information and the Material and Methods part was extended accordingly". However, I could not find any mention of neither tools in the supplementary information.

Reviewer #2:

Remarks to the Author:

The authors have addressed most of my comments and the manuscript is much improved. I have one major comment outstanding:

The authors have revised a statement to: "Our findings show that regimens including B. infantis NCDO 2203 supplementation (as compared with other probiotics, nutritional and antibiotic treatments) appear to have a highly beneficial effect on the early gastrointestinal microbiome, as reflected by centre K having the lowest NEC rates." (L465) and they note, the relation is now made on the overall regimen, not the probiotics itself.

I maintain my initial comment "This is strong statement and one wonders if the NEC rates significantly reduced after the probiotics were introduced into K. If not, it is misleading to relate the lower NEC rates specifically to the use of probiotics". If the authors are not able to show NEC rates have reduced significantly in K since the probiotics were introduced, they cannot make this claim.

Reviewer #3:

Remarks to the Author:

The authors here present an in-depth, systematic comparison of different NEC preventive regimens. The organization of the manuscript is clear, the scientific interpretations are comprehensive, and the clinical relevance is well stated. However, I have some major concerns regarding the statistical analyses performed in this paper that make it challenging to fully assess the robustness of the results and reproducibility. My major reservations are as follows:

Major comments:

1. Line 762. The network analysis in this paper was performed using Calypso. Its original paper indicates that "networks are generated by first computing associations between taxa using Pearson's correlation." Computing Pearson's correlation between compositional data is statistically invalid and could result in spurious correlations. This issue has been identified and studied in several papers. For instance:

1). Friedman, Jonathan, and Eric J. Alm. "Inferring correlation networks from genomic survey data." (2012): e1002687.

2). Lin, Huang, Merete Eggesbø, and Shyamal Das Peddada. "Linear and nonlinear correlation estimators unveil undescribed taxa interactions in microbiome data." Nature communications 13.1 (2022): 1-16.

Could authors clarify how these associations were calculated and whether the compositionality issues had been taken care of?

2. Line 764. The authors mentioned that the microbial composition analysis was performed using Microbiome Explorer package. This is a very unclear declaration since the package has many tuning parameters. For instance, there are two normalization methods included in the package: 1) calculating proportions and 2) using cumulative sum scaling (CSS). This first normalization method should not be chosen since it makes the compositionality issue worse, while the CSS method is compositionally robust. Similarly, there are four different differential abundance (DA) methods available in the package: DESeq2, Kruskal-Wallis, limma, as well as a zero-inflated log normal model, and DESeq2 should be the preferred method.

3. Lines 772 – 775. The compositionality issue applies to the correlation calculation as well. While the metabolomics data can be tricky to determine whether they are compositional or not, the microbiome relative abundance data are completely compositional. Computing the Pearson correlation between compositional data is statistically invalid. I suggest either 1) using compositionally robust methods, such as SparCC or SECOM, to compute correlations or 2) ALR/CLR/ILR transformed the relative abundances to get out of the simplex and then normalizing the transformed data to compute the correlations. Also, only stating bestNormalize function was implemented seems unclear. It is a wrapper function for many transformations, such as Lambert, Box Cox, etc. Stating which method did bestNormalize choose eventually would make the Methods section more transparent.

4. I share the concern of another reviewer regarding the batch effect. Although in lines 121-123, the authors have pointed out that the three hospitals are geographically close to each other, I wonder whether the batch effect (i.e., hospital) can be quantitatively evaluated. For instance, for the 16S or metagenomics data, PERMANOVA can be used to measure the proportion of R squared to each effect, so we can see how much variance is contributed by batch (technical variations), and how much is contributed by medication and probiotic regimens (biological variations).

5. Lines 174-175. "The clustering of G and L bacterial, archaeal, fungal and phagal microbiomes in the network analyses was somewhat unexpected, but indicated that extrinsic factors might have this strong separating effect." This sentence is a bit unclear. Firstly, the network is constructed based on differentially abundant genera. Based on Figures 1C and 1D, G and L tend to have similar trends for differentially abundant (archaeal and ascomycota/basidiomycota) genera as compared to K. Not sure if this is the same in other data. If so, the network clustering of G and L should be expected. Also, this sentence is a little confusing. Did the authors try to say that G and L are similar because of the same HM feeding and gentamicin prophylaxis, and these two are the leading effects in the clustering?

6. Lines 255-256. "In particular, the administration of Bifidobacterium and gentamicin correlated strongly with dissimilarities between the centres." Based on Figure 2E, it seems the administration of Bifidobacterium and gentamicin is almost perfectly linearly correlated. Any insights about it?

Minor comments:

1. Typos:

a) Line 112: what does "Star Methods" mean? Should it be "Methods"?

b) Line 112: faecal samples  fecal samples

2. Lines 116-117. Not sure if I missed anything, but it seems Table 2 does not contain the information about comparing metadata.

3. Lines 195-196. "Enterococcus was found to predominate the bacterial microbiome in G and L (77%), together with less abundant Lactobacillus and Staphylococcus"  "Enterococcus was found to predominate the bacterial microbiome in G and L (77%), followed by Lactobacillus and Staphylococcus"

REVIEWER COMMENTS and AUTHOR RESPONSES

Again, we appreciate the time of the reviewers and their valid contributions.

Please find our responses below.

Reviewer #1 (Remarks to the Author):

Overall, the authors have addressed most of the comments. Few additional minor comments below:

- The quality of the figures included in the pdf document with the main text has actually worsened compared to the original submission. The authors say they have submitted the files in the required 300dpi format, so the issue might only be in the inclusion of the figures in the main text document. Please make sure to fix this before publication.

We removed all Figures from the word file to avoid confusion. We upload them separately as foreseen in the submission process. We hope the quality of the figures is now suitable.

- "suppl. table 2" referred to in the main text does not exist anymore. Please include where appropriate the reference to suppl. table 2a and 2b.

We changed that accordingly.

- The authors replied: "*We included GUNC and CheckM2 accordingly to detect chimerism. The data can be found in the Supplementary Information and the Material and Methods part was extended accordingly*". However, I could not find any mention of neither tools in the supplementary information.

A table of GUNC, CheckM and CheckM2 outputs can be found in the supplementary information on github https://github.com/CharlotteJNeumann/preterm_shared/tree/main/Metagenomics/contigs. We have changed the wording accordingly to make it easier to understand.

No changes occurred when GUNC or CheckM2 was applied to the contigs that previously passed the contamination and completeness filters. For these contigs, contamination with GUNC is still ≤ 0.06 .

Reviewer #2 (Remarks to the Author):

The authors have addressed most of my comments and the manuscript is much improved. I have one major comment outstanding:

The authors have revised a statement to: *"Our findings show that regimens including B. infantis NCDO 2203 supplementation (as compared with other probiotics, nutritional and antibiotic treatments) appear to have a highly beneficial effect on the early gastrointestinal microbiome, as reflected by centre K having the lowest NEC rates."* (L465) and they note, the relation is now made on the overall regimen, not the probiotics itself.

I maintain my initial comment "This is strong statement and one wonders if the NEC rates significantly reduced after the probiotics were introduced into K. If not, it is misleading to relate the lower NEC rates specifically to the use of probiotics". If the authors are not able to show NEC rates have reduced significantly in K since the probiotics were introduced, they cannot make this claim.

We deleted that statement.

Reviewer #3 (Remarks to the Author):

The authors here present an in-depth, systematic comparison of different NEC preventive regimens. The organization of the manuscript is clear, the scientific interpretations are comprehensive, and the clinical relevance is well stated. However, I have some major concerns regarding the statistical analyses performed in this paper that make it challenging to fully assess the robustness of the results and reproducibility. My major reservations are as follows:

Major comments:

1. Line 762. The network analysis in this paper was performed using Calypso. Its original paper indicates that "networks are generated by first computing associations between taxa using Pearson's correlation." Computing Pearson's correlation between compositional data is statistically invalid and could result in spurious correlations. This issue has been identified and studied in several papers. For instance:

- 1). Friedman, Jonathan, and Eric J. Alm. "Inferring correlation networks from genomic survey data." (2012): e1002687.
- 2). Lin, Huang, Merete Eggesbø, and Shyamal Das Peddada. "Linear and nonlinear correlation estimators unveil undescribed taxa interactions in microbiome data." Nature communications 13.1 (2022): 1-16.

Could authors clarify how these associations were calculated and whether the compositionality issues had been taken care of?

We changed the network accordingly, using SparCC within the SCNIC tool. We adapted the M&M part and the results section accordingly. Nevertheless, we decided to move the network to the supplementary information- it can now be found in Suppl. Fig. 2.

2. Line 764. The authors mentioned that the microbial composition analysis was performed using Microbiome Explorer package. This is a very unclear declaration since the package has many tuning parameters. For instance, there are two normalization methods included in the package: 1) calculating proportions and 2) using cumulative sum scaling (CSS). This first normalization method should not be chosen since it makes the compositionality issue worse, while the CSS method is compositionally robust. Similarly, there are four different differential abundance (DA) methods available in the package: DESeq2, Kruskal-Wallis, limma, as well as a zero-inflated log normal model, and DESeq2 should be the preferred method.

We performed CSS and DESeq2 . This is now clearly stated in the Materials and Methods which have been adapted wherever necessary.

Wherever necessary, we changed the figures and descriptions accordingly. Asterisks in the boxplots are now referring to the DESeq2 values.

The DESeq2 tables are available on Github.

3. Lines 772 – 775. Heatmap

The compositionality issue applies to the correlation calculation as well. While the metabolomics data can be tricky to determine whether they are compositional or not, the microbiome relative abundance data are completely compositional. Computing the Pearson correlation between compositional data is statistically invalid. I suggest either 1) using compositionally robust methods, such as SparCC or SECOM, to compute correlations or 2) ALR/CLR/ILR transformed the relative abundances to get out of the simplex and then normalizing the transformed data to compute the correlations. Also, only stating bestNormalize function was implemented seems unclear. It is a wrapper function for many transformations, such as Lambert, Box Cox, etc. Stating which method did bestNormalize choose eventually would make the Methods section more transparent.

We corrected this accordingly to your suggestion of applying CLR transformation and included this section into the M&M.

We changed the heatmap accordingly, as well as parts in the results section which referred to the data of the old heatmap.

The list of methods bestNormalize chose per genus and metabolite is now available on github and we added the respective part in the M&M section.

4. I share the concern of another reviewer regarding the batch effect. Although in lines 121-123, the authors have pointed out that the three hospitals are geographically close to each other, I wonder whether the batch effect (i.e., hospital) can be quantitatively evaluated. For instance, for the 16S or metagenomics data, PERMANOVA can be used to measure the proportion of R squared to each effect, so we can see how much variance is contributed by batch (technical variations), and how much is contributed by medication and probiotic regimens (biological variations).

We performed PERMANOVA as requested, and the results are provided in Supplementary Table 1. Results were integrated in lines 122 ff and 259 ff.

Indeed we could show that the variance explained by the regimen (nutrition, Bifidobacterium administration, antibiotics) is the same (all samples) or even higher (tp3, 7) than by the hospital.

5. Lines 174-175. "*The clustering of G and L bacterial, archaeal, fungal and phagal microbiomes in the network analyses was somewhat unexpected, but indicated that extrinsic factors might have this strong separating effect.*" This sentence is a bit unclear. Firstly, the network is constructed based on differentially abundant genera. Based on Figures 1C and 1D, G and L tend to have similar trends for differentially abundant (archaeal and ascomycota/basidiomycota) genera as compared to K. Not sure if this is the same in other data. If so, the network clustering of G and L should be expected. Also, this sentence is a little confusing. Did the authors try to say that G and L are similar because of the same HM feeding and gentamicin prophylaxis, and these two are the leading effects in the clustering?

We have deleted this sentence and instead referred to the Permanova analysis, which underscores what we wanted to express here.

6. Lines 255-256. "In particular, the administration of Bifidobacterium and gentamicin correlated strongly with dissimilarities between the centres." Based on Figure 2E, it seems the administration of Bifidobacterium and gentamicin is almost perfectly linearly correlated. Any insights about it?

This correlation appears, as the absence of the one means the presence of the other (Gentamycin but no Bifidobacterium in G & L; Bifidobacterium but no Gentamycin in K) and vice versa.

Minor comments:

1. Typos:

a) Line 112: what does "Star Methods" mean? Should it be "Methods"?

Nature communications requests the naming of "Star Methods" for the Methods section.

b) Line 112: faecal samples  fecal samples

This was corrected (also in the abstract)

2. Lines 116-117. Not sure if I missed anything, but it seems Table 2 does not contain the information about comparing metadata.

This refers to table 2 in the named reference.

3. Lines 195-196. "Enterococcus was found to predominate the bacterial microbiome in G and L (77%), together with less abundant Lactobacillus and Staphylococcus"  "Enterococcus was found to predominate the bacterial microbiome in G and L (77%), followed by Lactobacillus and Staphylococcus"

This was corrected.

REVIEWERS' COMMENTS

Reviewer #3 (Remarks to the Author):

Most of my comments were addressed appropriately, and the paper was consistently improved.

Here is one minor comment on Fig. 1:

It seems the top X genera in Fig. 1C and 1D are defined by the rank of average relative abundances across centers (or perhaps something else). Please state the definition of "top X genera" in the figure legend accordingly.

Response to the reviewers:

Reviewer #3 (Remarks to the Author):

Most of my comments were addressed appropriately, and the paper was consistently improved.

Here is one minor comment on Fig. 1:

It seems the top X genera in Fig. 1C and 1D are defined by the rank of average relative abundances across centers (or perhaps something else). Please state the definition of "top X genera" in the figure legend accordingly.

Response:

The legend was corrected and information was added.